collaborative care; global mental health; global mental health policy; prevention strategies; suicide

**Corresponding author:**
Nikhil Jain;
Email: n.jain@maastrichtuniversity.nl

# Role of law enforcement agencies in suicide prevention: A scoping review

Nikhil Jain[1,2] , Isha Lohumi[2] , Niranjana Regimon[2], Aratrika Datta[2]  and Soumitra Pathare[2] 

[1]Maastricht University Faculty of Health Medicine and Life Sciences, Netherlands and [2]Centre for Mental Health Law & Policy, Indian Law Society, India

## Abstract

The role of Law enforcement agencies (LEA) is significant in suicide prevention efforts as first responders. Nevertheless, no published study to date has systematically compiled the body of knowledge about suicide prevention efforts involving LEA. The current scoping review aims to methodically map and examine the peer-reviewed literature and grey literature on the role of LEA in suicide prevention. Electronic searches of the databases like Medline, PsycINFO, Google Scholar, Web of Science, Scopus, CINAHL and Google were conducted using a comprehensive search strategy to identify relevant resources. Grey literature was searched searches were undertaken on relevant databases and, as well as government and organisational websites. The reporting of the review followed the Preferred Reporting Items for Systematic Reviews and Meta-Analyses Extension for Scoping Reviews guidelines. The inclusion criteria comprised research articles, reports, and guidelines/policy documents on the role of law enforcement agencies (LEAs) in suicide prevention. Studies on prevalence, custodial settings, non-English publications, and reviews were excluded. Inclusion criteria comprised research articles, reports and guidelines/policy documents focusing on law enforcement's role in suicide prevention. Studies focusing solely on prevalence or epidemiology, studies confined strictly to custodial settings, publications not in English and systematic reviews or meta-analyses were excluded. Out of 3,327 records screened, the full texts of 82 resources were included in the review. All the resources identified were categorised between peer-reviewed literature and grey literature. Resources were thematically categorised based on functional roles into- I. Strategic and System-Embedded Roles of LEA, II. Capacity Building and Training Oriented Engagements, III. Surveillance Reporting and Data Systems Role, IV. Community Facing and Preventive Engagement, and V. Means Restriction and Environmental Prevention Roles. The chronological development of the resources was inconsistent. Most resources were from high-income countries, focusing on the evaluation of training, capacity building programmes, surveillance initiatives and the exploration of varied roles of LEA across custodial, community and crisis settings and other interdisciplinary collaborations. Notably, the resources show increased disparity in quantity and research methodological approaches across geographies. The review highlights substantial heterogeneity and a limited resource base from low- and middle-income countries on the role of LEA in suicide prevention, with a dearth of structured, evidence-based, scalable models in these settings. These gaps point to an urgent need for locally relevant and cross-sectoral models that position law enforcement as integral partners in suicide prevention efforts, especially where these agencies play a major role as first responders.

## Impact statement

This scoping review maps and examines the current research evidence on the role of law enforcement in suicide prevention. It traces the development of their engagement from a narrow, legalistic frame of suicide to an increasing recognition of law enforcement as an important partner of multisectoral, collaborative responses to this public health problem. The review also points to major geographical inequalities in the published research, calling for efforts to build community-focused and integrated contributions of the law enforcement agencies in low- and middle-income countries (LMICs), where the heavy burden of suicide, large treatment gaps and intricate determinants of suicide require their crucial engagement.

## Introduction

World Health Organisation's (WHO) Comprehensive Mental Health Action Plan focuses on enhancing responses to self-harm and suicide by engaging and training non-health sectors, such as law enforcement, in assessment, support provision and follow-up of individuals at risk of suicide

(World Health Organisation, 2013). Law enforcement agency (LEA) is a term conventionally used to describe police and related authorities involved in upholding and enforcing the law as well as maintaining public safety. These are uniquely situated to support this cause within the global agenda to reduce increasing suicide rates, viewing suicide as a public health issue. In sustainable development goals (SDGs), SDG-3 (good health and well-being) and SDG-16 (peace, justice and strong institutions) emphasise LEAs' responsibilities in engaging with individuals in crisis, safeguarding communities and playing roles in prevention and postvention efforts. SDG-3 target 3.4.2 aims to reduce suicide mortality by one-third by 2030, while targets 16.1, 16.2 and 16.3 of SDG-16 specify the need for accountable and transparent policing that enforces equitable protection and access to justice for marginalised and vulnerable groups (United Nations, 2015a, 2015b). Although they play a pivotal role, the evidence base for LEAs in preventing suicide is scarce and limited.

Much of the previous research in this field concentrated narrowly on the investigative functions or custodial responsibilities (Best et al., 2006), within prisons or other such institutional settings. Although the contribution of LEA in safeguarding vulnerable groups in this context is instrumental, it presents only a fraction of their possible contribution. Their wider participation within community settings for various interventions targeted at early detection, promotion and prevention is underexplored (Oyama and Sakashita, 2017; Khorasheh et al., 2019; Spagnolo and Lal, 2021; Morgan et al., 2022). Most of the earlier studies highlighted specific functions of the LEA such as crisis intervention (Khorasheh et al., 2019; Spagnolo and Lal, 2021), first response (Canada et al., 2010; Burnette et al., 2015; Arensman et al., 2016; Oyama and Sakashita, 2017; Chidgey et al., 2019; Morgan et al., 2022), (National Institute for Health and Care Excellence, 2018), gatekeeping (Canada et al., 2010; Burnette et al., 2015; Arensman et al., 2016; Oyama and Sakashita, 2017; Chidgey et al., 2019; Morgan et al., 2022) and handling suicide notifications (Department of Health, 2019; Thorne and O'Reilly, 2022). Other functions included patrolling high-risk locations like bridges or public transport hubs and collecting data (Pirkis et al., 2015; Public Health England, 2015; National Institute for Health and Care Excellence, 2018), such as real-time suicide surveillance (Baran et al., 2021; Krishnamoorthy et al., 2023) or on institutional risks and support programmes for LEA themselves (Pirkis et al., 2015; Public Health England, 2015; National Institute for Health and Care Excellence, 2018; Struszczyk et al., 2019; Johnson et al., 2022). In other cases, their contribution is conceptualised in a very limited scope as a partner with other societal institutions or organisations (Matheson et al., 2005; Chidgey et al., 2019). However, these roles are often defined in the context of being situational and not systematic. Moreover, there remains a significant gap in evidence concerning LEA-led or supported wider preventive or promotive community-based programmes (Norris and Cooke, 2000; Arensman et al., 2016; londoño et al., 2020; Krishnamoorthy et al., 2023, 2025).

Although sporadic evidence is available across custodial, investigative, crisis response and self-care approaches, the evidence base is fragmented. It draws attention to an important gap, particularly in understanding how LEA can be strategically included within national and international suicide prevention strategies and other policy-driven initiatives. This review aims to fill the gap through the following objectives: 1) to map and examine existing literature on suicide prevention interventions involving LEA and 2) to systematically scope existing policies, guidelines and strategies that address suicide prevention approaches involving LEA.

## Methods

The review follows the Preferred Reporting Items for Systematic Reviews and Meta-Analyses Extension for Scoping Reviews (PRISMA-ScR) guidelines (Tricco et al., 2018).

### Search strategy and selection criteria

A search strategy was created using the following key research concepts: (1) law enforcement personnel or police officers (population and context) and (2) suicide prevention (concept and outcome) (refer to detailed search strategy in the Supplementary Material). The reviewers (AD and NR) searched seven electronic databases, Embase, PubMed, PsycINFO, Web of Science, CINAHL and Scopus, using the search terms "law enforcement personnel*" AND "(suicide OR attempted suicide)" to identify studies on the role of LEA in suicide prevention. The search terms for each concept were combined using Boolean operators. The search was run between July 2024 and December 2025. Database searches were supplemented by a search of grey and unpublished literature and were explored using keywords and phrases from published material (AD and NR). The same keywords were used for searches on Google and Google Scholar, and the first 1,000 results were reviewed, respectively, as recommended in various reviews (Godin et al., 2015; Handerer et al., 2022). The sources of grey literature were limited to documents that were publicly accessible and part of the search results on Google and Google Scholar. Reference lists of past reviews were also explored (by AD and NR) to find other studies that were included based on the inclusion criteria, as proposed in some reviews, to ensure the search was exhaustive and expansive (Godin et al., 2015; londoño et al., 2020; Handerer et al., 2022).

The research incorporated peer-reviewed articles, reports and guidelines/policy documents written in English that suggested, depicted, assessed or analysed interventions, laid out policies or practices initiated by LEA for suicide prevention. For inclusion, research had to be centred on LEA participation in suicide prevention, either as a main topic or a significant part. There were no geographical limitations, and studies from any country or region could be included. Both peer-reviewed papers and grey literature, i.e., government reports, organisational policies and policy documents, were eligible. Studies were not considered if they were not published in English (due to practical limitations), examined suicide prevention strategies targeted at LEA, strictly confined to the prison environment, or where LEA's role was incidental or undefined. No date restrictions were applied, as the aim was to map evidence from all available years. Systematic reviews and meta-analyses were also excluded to focus on primary studies with detailed evidence on the role of law enforcement in suicide prevention pathways that are not accessible through aggregated syntheses.

### Screening and selection

The search results from each database were imported into Rayyan (Ouzzani et al., 2016), for the screening and de-duplication of the publications (by AD and NR). Two independent researchers (AD and NR) screened all identified articles by title and abstract to exclude papers that did not meet the inclusion criteria. De-duplication and initial screening were conducted in Rayyan, with Group 1 (AD, IL) and Group 2 (NR, NJ) screening independently. Any discrepancies within

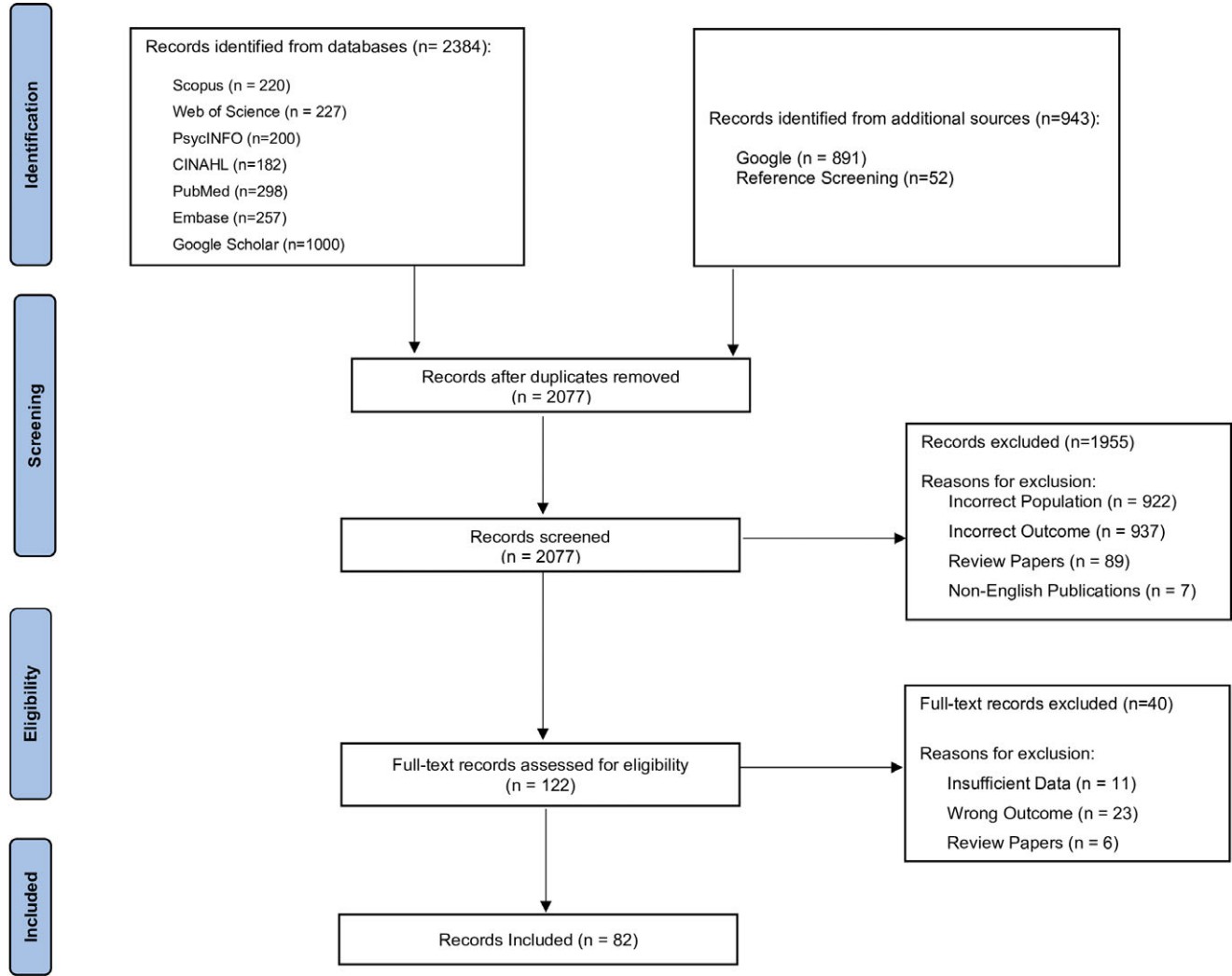

**Figure 1.** PRISMA Chart.

groups were resolved through discussion to reach consensus. If consensus could not be reached within a group, the issue was escalated to a cross-group discussion. Persistent disagreements were resolved by involving a third reviewer (SP) to make the final decision on study inclusion. The screening process and reasons for exclusion and inclusion are presented in the PRISMA chart as Figure 1.

## Data extraction

The following data were extracted into an MS Excel (Microsoft Cooperation, 2024) spreadsheet (by AD and NR): title, authors, country, year of publication, type of literary resource, objective, study design, sample description, outcomes and recommended role of law enforcement agencies. For resources describing or evaluating an intervention, additional data were extracted on intervention description, duration, mode of delivery and outcome measurement tool. Data extracted from grey literature included title, year, country, region, author, geographical scope, scope or objective and key roles and responsibilities of LEAs described in the document, ensuring consistency in narrative synthesis and comparability across evidence types.

## Data analysis

A narrative synthesis summarised the findings, mapped the scope and nature of included studies and identified patterns in LEA involvement in suicide prevention (Godin et al., 2015; Handerer et al., 2022). Literary resources were classified as peer-reviewed sources and grey literature. Peer-reviewed sources focusing on the implementation or evaluation of LEA interventions or describing existing mechanisms and processes without reporting intervention results, were included in the study. Grey literature included international guidance outlining overarching aims, national and state policies articulating country or state-specific visions and action plans and country-specific guidance documents recommending practice approaches to suicide prevention.

## Results

The study identified 3,327 resources through selected databases and grey literature searches, and an additional 54 through reference screening of the identified resources. After de-duplication, 2077 resources were included for further screening. We excluded 922 resources that did not focus on LEA, 937 that did not address suicide prevention specifically, 89 systematic or scoping reviews

and 7 non-English publications, leaving 122 for full-text review. During full-text screening, we excluded 6 review papers, 11 resources with insufficient data and 23 resources that examined suicides within LEA and other outcomes outside the scope of this study. At last, a total of 82 resources were included in this review, describing intervention-based trials targeted at capacity-building and training of law enforcement agencies and community suicide prevention models, interdepartmental collaboration and other strategies adopted by or recommended for suicide prevention targeted at law enforcement.

### Evidence characteristics of the identified resources

The evidence literature comprised peer-reviewed studies ($n = 51$) and grey literature ($n = 31$). The scope of the resources varied widely, addressing critical areas such as identifying the role of law enforcement agencies in suicide prevention (McGeechan et al., 2017; White and Hussain, 2020), training and intervention strategies involving LEAs for suicide prevention (World Health, 2012; Norton, 2017; White and Hussain, 2020); the effectiveness of suicide prevention interventions involving LEAs (Lee et al., 2015; Brooks-Russell et al., 2019) and the importance of LEAs in postvention efforts (Ko et al., 2021).

### Chronological mapping of the literature

From 1961 to 1970, and 1991–2000, four peer-reviewed studies formed the evidence base, while between 2001and 2010, two peer-reviewed studies explored LEA contact with individuals before suicide, and two sources of grey literature related to LEAs' role within the suicide prevention space formed the evidence base from that period.

Further, between 2011 and 2020, 34 sources were published, comprising 21 peer-reviewed literature and 13 grey literature. Additionally, 39 publications were included from 2021 onwards, comprising 24 peer-reviewed and 15 grey literature, mostly evaluation studies of LEA training programmes, joint LEA–mental health initiatives and other multi-sectoral strategies (Figure 2).

### Regional distribution of the literature

The review identified a diverse range of resources concerning suicide prevention, spanning various regions and income groups, as per the World Bank Classification for the 2025 fiscal year. Of the 82 studies included, a significant portion originated from high-income countries ($n = 57$), including the United States, the United Kingdom, Australia, the Netherlands, Ireland, Portugal, Germany, South Korea, Sweden, Guyana, New Zealand and Canada (refer to Figure 3). Additionally, resources from upper-middle-income countries ($n = 3$), such as Malaysia, South Africa and China and low- and middle-income countries ($n = 19$), such as India, Pakistan, Bhutan, Nepal and Nigeria were also included. The review also included three global-level resources ($n = 3$) that were not specific to any single country (refer to Figure 3).

### Functional domain of law enforcement engagement in suicide prevention

This section maps how the literature conceptualises the roles and responsibilities of law enforcement agencies in suicide prevention, drawing on global institutional, state or national level policies, strategies and guidance documents and empirical studies (Refer to Table 1, Table 2, and Table 3).

#### Strategic and system-embedded roles of law enforcement agencies

This section synthesises evidence from 37 sources (20 grey literature and 17 peer-reviewed studies) examining the strategic and system-embedded roles of law enforcement agencies within suicide prevention frameworks.

Included sources describe LEAs as system-embedded actors in suicide prevention, with roles defined through constitutional mandates, statutory provisions and human rights–based obligations. Across jurisdictions, legal frameworks establish a duty of care requiring LEAs to take mandatory preventive measures to protect individuals at risk of suicide, while they are in custody (National Human Rights Commission India, 2014; Department of Health,

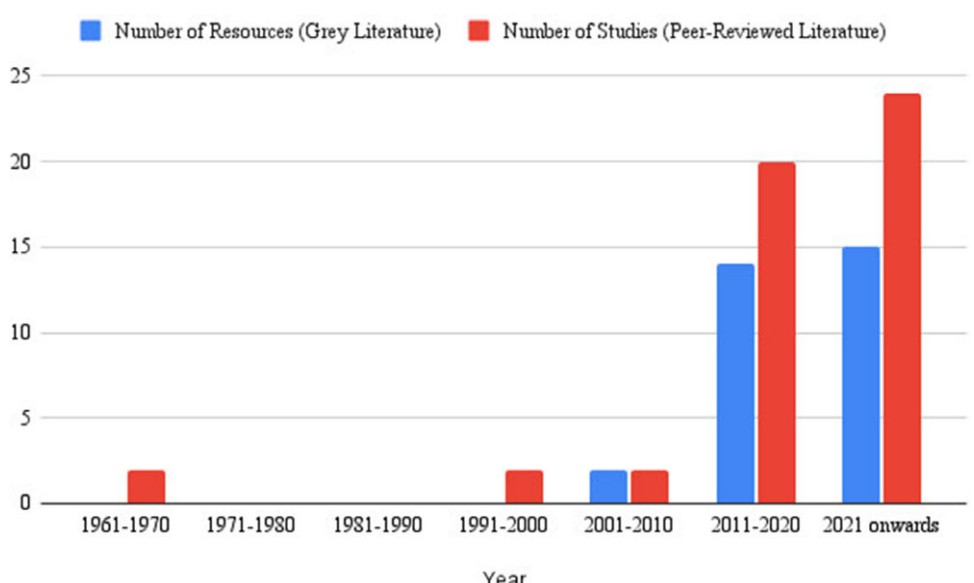

**Figure 2.** Chronological Development of Resources.

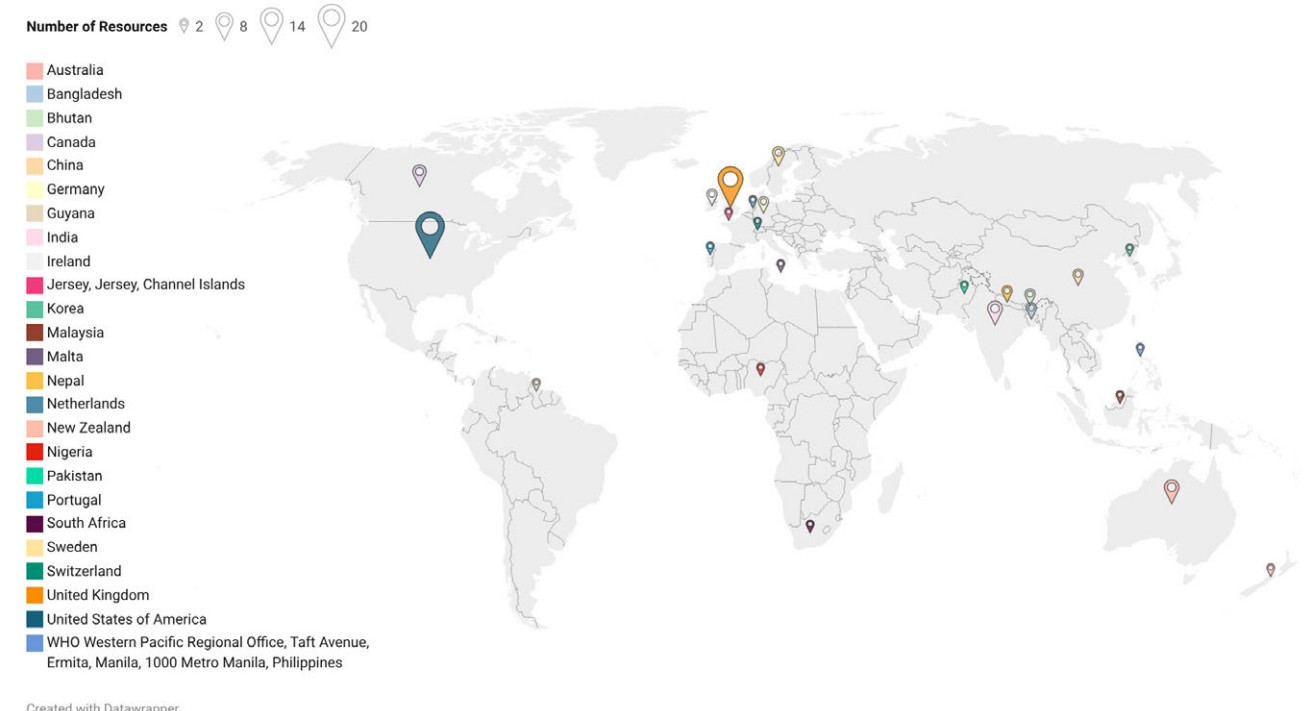

**Figure 3.** Geographical Distribution of Resources.

2019). These duties apply to both acts and omissions and is framed as a foreseeable and preventable harm.

Most of the national and sub-national suicide prevention strategies position LEAs as frontline responders to suicide. Stipulated roles include early identification of suicide risk, scene safety management, restriction of access to lethal means, preliminary risk assessment and facilitation of referral or transfer to mental health services. They are primarily framed as protective and facilitative agents (World Health Organization, 2009; Shrestha et al., 2021; Hill et al., 2022). LEAs are delegated functionally limited authority to detain, to enable emergency mental health evaluation, with explicit guidance, in most policy and guidance documents, to avoid undue criminalisation or investigative framing of suicidal behaviour (Bouveng et al., 2017; Royal Government of Bhutan, 2018; Shaw, 2021). Across the identified resources, intersectoral coordination was framed as a facilitative function of LEAs, enabling coordination between emergency, health, justice and community-based services rather than exercising directive control.

Some of the identified resources used suicide prevention frameworks to divide functional roles of LEAs across prevention (population level), intervention (acute crisis response) and postvention (post suicide or attempt mitigation) as per the temporal risk continuum of suicidal behaviour (Hadlaczky et al., 2016; Sher, 2016; Norton, 2017; The Youth Suicide Prevention Life Promotion Collaborative, 2023). These include participation in community prevention activities, emergency response to suicidal crises, coordination with health and social welfare systems, support to bereaved families and contribution to suicide surveillance, review and monitoring processes.

Out of the 17 peer-reviewed studies describing LEA integration within broader suicide prevention, crisis response and public health systems, most used quantitative or mixed-methods analysis focused on role conceptualisation, system positioning and cross-sector alignment. Indicators were predominantly organisational and system-level, including establishment of specialised units, integration of police functions within national or regional strategies, formalised crisis response and postvention roles and alignment with health and emergency service protocols. Strategic functions such as early detection, coordination during high-risk incidents, crisis negotiation and facilitation of access to care were commonly described, while effectiveness was inferred mainly from descriptive accounts or stakeholder perspectives rather than comparative or longitudinal analyses.

### Capacity building and training-oriented engagements

Out of the 30 identified resources describing capacity building and training, oriented engagements, 24 were peer-reviewed publications and 11 were grey literature sources. Across the included studies, the gatekeeper framework ($n = 28$) positioned LEAs in a non-clinical role within suicide prevention systems, emphasising their function as first responders responsible for recognising warning signs, assessing risk level and facilitating referral to appropriate professional services.

Ten sources explicitly used the term gatekeeper training, while 14 sources described similar ingredients that could build their capacity as gatekeepers.

Of the 24 resources addressing gatekeeper approaches, 18 described LEA capacity building at a universal level, situating gatekeeper training within broader health system strengthening efforts, while 6 focused on capacity building to respond to specific populations or contexts, including youth, individuals with frequent LEA contact and inter-agency first responder collaborations. Gatekeeper training models varied substantially in format and intensity, ranging from brief sessions (approximately 90 min) to multi-hour curricula incorporating skills rehearsal and applied learning components (Arensman et al., 2016; Osteen et al., 2021). Across studies, training was associated with improvements in suicide-related knowledge, attitudes and self-reported confidence in responding

**Table 1.** Evidence characteristics of peer-reviewed literature

| References | Objective | Study design | Sample description | Results | Role of LEA | Thematic synthesis |
|---|---|---|---|---|---|---|
| McGee (1968) | To demonstrate how operational suicide prevention programmes in Florida exemplify and operationalise core principles of community mental health; | NA | professional consultants, including mental health professionals, a directorate body, community advisory boards including sheriffs, police chiefs, hospital administrators, etc. and centre staff, including research associates and clinical staff | NA | Describes the need for representatives from various sectors, including law enforcement, in the community advisory board in the organisational model of the community suicide prevention programme to affect the policies and procedures that enable the suicide prevention service and the other existing agencies to make maximum use of one another | Intersectoral Collaboration and Community Engagement |
| Roberts and Joseph (1970) | To learn the specific activities of the suicide prevention agencies that were directed at crisis intervention, the relationship of these activities to the prevention of suicide, how the emergency telephone service operated, what initial procedures were used in crisis intervention and the kinds of resources to which clients were referred. | Primary Research (cross-sectional) | 24 suicide prevention agencies of the Standard 212 Metropolitan Statistical Areas of the United States, | Agencies used trained volunteers and professional staff; no written plan for handling callers at risk, but they had operational principles to guide them. A pattern for initial intervention emerged: establishment of rapport, evaluation of the caller's suicidal potential and deciding on the course of action to be taken. Most calls are handled by telephone counselling alone; limited evidence of effectiveness; 24-h availability varies; follow-up studies of emergency callers are needed to determine agency effectiveness | Police and emergency services were part of the referral network for suicide prevention agencies; agencies coordinated with police for emergencies; police involvement was mentioned as part of a multi-agency crisis response system | Intersectoral Collaboration and Community Engagement, Strategic and System-Embedded Roles of Law Enforcement Agencies |
| Olivero and Hansen (1994) | To analyse the linkage agreement between law enforcement agencies and community mental health services in the management of persons experiencing suicidal episodes. | | Qualitative: 15 emergency community mental health workers, 45 police officers and/or police chiefs, representing every police and sheriff's department in the two-county area and a 6-week, on-site participatory observation study of the programme in the summer of 1989. Quantitative: 320 case report forms were analysed for the year 1987. (Secondary data) | 1. police officers utilised consultations with mental health providers when confronted with persons experiencing suicidal episodes. 38.3% of the consultations between police and mental health workers concerned individuals suffering suicidal episodes, followed by those involving persons who were psychotic and acting out. 2. The police agencies serving populations of similar size utilised services at a similar rate, with the rate of utilisation of services increasing with the size of population served. 3. In 81.7% cases, police came across persons experiencing suicidal episodes when there was no criminal charge involved. 4. 43.5% of the clients had a history of services (substance abuse or mental health services) with MHSFWCI. 5. The most common crisis resolution method was to release the individual to relatives after scheduling an appointment to return for services at MHSFWCI. In other instances, the emergency worker took immediate measures to ensure the individual's safety | Generated evidence on utilisation of mental health services by police; however, their assessment for identifying the risk was questionable due to a lack of formal training. Reaffirmed the role of LEA as first responders in suicide related events in community setting highlighting the associated challenges in managing the crisis, such as transportation to the facility. | Intersectoral Collaboration and Community Engagement, Capacity Building and Training-Oriented Engagements |

*(Continued)*

*Cambridge Prisms: Global Mental Health*

| References | Objective | Study design | Sample description | Results | Role of LEA | Thematic synthesis |
|---|---|---|---|---|---|---|
| Potter et al. (1995) | To describe the public health approach to suicide prevention, outlining its four core components and demonstrate how this approach is applied | NA | NA | NA | The gatekeeper programme provides training to community members, such as police, to help identify youth at risk of suicide and refer them as appropriate. | Capacity Building and Training-Oriented Engagements |
| Matheson et al. (2005) | to examine the epidemiology of calls for police assistance in relation to suicide and suicide attempts in an urban environment and to identify implications for police training and preparedness. | Retrospective Cross-sectional | NA | During the five years, there were an average of 1,422 annual calls for suicidal behaviour, 631 involving men and 791 involving women. Completed suicides accounted for 15% of calls (24% of male callers and 8% of female callers). Over time, calls for assistance for incidents of suicidal behaviour increased by 11% overall (an increase of 4% in the number of calls involving men and of 17% in the number of calls involving women). The number of completed suicides, over the five-year study period, each division employed an average of 188 police officers, who responded to241,253 dispatched calls per year. These numbers translate to 259 calls per officer in each division per year. Decreased by 24% over the study period (22% male callers and 31% female callers). | Police assistance in suicide prevention, police records of suicide data, police training initiatives on understanding the changing patterns of suicidal behaviour | Capacity Building and Training-Oriented Engagements, Strategic and System-Embedded Roles of Law Enforcement Agencies, Surveillance, Reporting and Data Systems Role |
| Linsley et al. (2007) | To evaluate police-contact with individuals before suicide and associated health service contact | Secondary (retrospective) | 205 death records from the mortality registers and the coroner's office | Out of the 205 suicide cases, 24 individuals (12%) had contact with the police within 3 months of death as victims of crime and 24 individuals (12%) had been arrested as alleged perpetrators of crime. Seven individuals had been both a victim of crime and an alleged perpetrator in the 3 months, leaving an actual total of 41 people (20%) who had been in contact with the police. Among the 41 cases with police contact, 17 (41%) had also seen their general practitioner (GP) within the same period. In addition to the main findings, 21 cases had impending court appearances (for criminal matters), making it likely that the individual had ongoing police contact. Of these, 14 had been arrested within the last 3 months, and of these 14 people, 6 had also reported a crime within that period | Local protocols between health agencies, especially mental health services and police, should be established. This has to be backed up by training programmes for police on recognising suicide risk factors and dealing with suicidal individuals effectively. | Intersectoral collaboration and Community Engagement, Capacity Building and Training-Oriented Engagements |
| Schlebusch (2012) | To identify priorities and prevention strategies for reducing suicidal behaviour in South Africa by discussing a framework for a proposed national prevention programme. | NA | NA | NA | Further education and training programmes for those who encounter suicidal individuals: Promoting skills development on how to support and assist suicidal patients and their families in obtaining help | Capacity Building and Training-Oriented Engagements |

**Table 1.** (*Continued*)

| References | Objective | Study design | Sample description | Results | Role of LEA | Thematic synthesis |
|---|---|---|---|---|---|---|
| Draper et al. (2015) | The paper presents A sample of the research and rationale that underpinned the development of the policy for Helping Callers at Imminent Risk of Suicide | | NA | NA | Central emergency responders for active rescue; formal collaboration with crisis centres; information-sharing protocols with 911; promote least-invasive interventions (mobile crisis teams) with police as escalation; cross-training with crisis services; safety confirmation and follow-up protocols | Intersectoral Collaboration and Community Engagement, Strategic and System-Embedded Roles of Law Enforcement Agencies |
| Hadlaczky et al. (2016) | To present two commonly used evidence-based frameworks for classifying preventive strategies applied in public health: (a) the "primary, secondary, tertiary" prevention model, and (b) the United States Institute of Medicine model, which incorporates universal, selective and indicated approaches. To illustrate the utility of such models, examples of current applications of suicide prevention activities and programmes from across the globe are examined. | NA | NA | NA | Police are identified as key gatekeepers and community facilitators within the primary, secondary and tertiary prevention model services for individuals in acute distress. | Strategic and System-Embedded Roles of Law Enforcement Agencies |
| Hagaman et al. (2016) | To explore the perceptions, practices and politicisation of suicide reporting and to analyse networks drawn by police, policy and health officials to better understand vital surveillance in Nepal and investigate how institutional networks affect how suicide deaths might be (un) documented and (un) reported within varying institutions. | Descriptive Study | Ethnographic fieldwork including participant observation in health, development and legal institutions and 36 semi-structured interviews were conducted across law and health system levels including multilateral organisations (WHO, World Bank, Institute of Migration), foreign aid agencies (DfID; USAID), government ministries (Ministry of Health and Population, Ministry of Home Affairs), healthcare institutions (government hospitals, academic hospitals), legal and law enforcement institutions (district police, national police academy) and | A discrepancy between perceived criminality of suicide and actual legal codes; The dominant role of police in collecting information, reporting suicide and interacting with families affected by suicide; A lack of systematic, nationally standardised approaches within the health system for documentation and reporting of suicide, including limited communication channels between HMIS and global reporting (e.g., through WHO) of suicide statistics; and Limited engagement of families in reporting suicide because of fear of legal entanglements anticipated with reporting suicide, anticipated stigma for families of suicide victims and greater time and financial burden compared to reporting natural deaths. | Police, as the largest contributor to documenting suicide data, are the majority of health informants and are involved in the confirmation, investigation and communication with the family and victims are mainly conducted through the police. Education of police involved in suicide death documentation and communication with families, myths and subsequent stigma may be dispelled. Collaborative, multi-sectoral approaches, especially partnerships between law enforcement and the health system, are needed for reliable and accurate surveillance and | Surveillance, Reporting and Data Systems Role, Intersectoral Collaboration and Community Engagement, Capacity Building and Training-Oriented Engagements |

(*Continued*)

| References | Objective | Study design | Sample description | Results | Role of LEA | Thematic synthesis |
|---|---|---|---|---|---|---|
| | | | nongovernmental organisations (mental health and psychosocial organisations, advocacy organisations) | | ultimately for effective suicide prevention | |
| Sher (2016) | Synthesise research on the police's dual role in suicide prevention, both direct (training, intervention with suicidal persons) and indirect (drug law enforcement, community outreach, reducing risk factors) | NA | NA | NA | Direct Prevention: gatekeeper functions, mental health recognition Indirect Prevention: Drug law enforcement (reduces drug availability) + community outreach (reduces antisocial behaviour—a suicide predictor) | Capacity Building and Training-Oriented Engagements, Strategic and System-Embedded Roles of Law Enforcement Agencies |
| Dorji et al. (2017) | This article documents Bhutan's policy and governance for addressing depression and suicide within the context of its national suicide-prevention strategy, examines progress and highlights lessons for future directions in suicide prevention | | NA | NA | Police identified as key sector partner in suicide prevention; included in multi-sectoral platform of stakeholders under government leadership; participate in capacity-building activities as gatekeepers; contribute to improved suicide information systems to inform policy and decision-making | Strategic and System-Embedded Roles of Law Enforcement Agencies, Surveillance, Reporting and Data Systems Role, Intersectoral Collaboration and Community Engagement |
| Norton (2017) | To map out the best practices in suicide prevention through postvention by identifying the distinct roles and functions of multiple first responder communities and to guide how first responders can reduce risk and promote healing for individuals newly bereaved by suicide. | NA | First responders, including law enforcement, emergency medical services, coroners or medical examiners and funeral directors | This article establishes a comprehensive framework for first responder response to suicide death as a distinct phase of suicide prevention (postvention), separate from crisis intervention with suicidal individuals. For postvention response to a suicide, it is important to broaden the usual law enforcement and emergency medical services (EMS) definition of first responders to also include medical examiners/coroners, funeral directors, media and faith leaders. Despite the important roles they play, most first responders have had no formal training in postvention and, subsequently, are often not familiar with best practices for an effective response. Training would increase understanding about best practices and the importance of their roles, sensitivity to the needs of immediate family and the broader community after a suicide death. It also provides them with shared concepts and language necessary for an integrated and effective. | provision of practical assistance to loss survivors by police, assisting with cleaning and restoration of the death scene, reassurance and support while providing a suicide death notification to loved ones, law enforcement training for death notification, self-care and training for postvention response | Capacity Building and Training-Oriented Engagements, Strategic and System-Embedded Roles of Law Enforcement Agencies |

| References | Objective | Study design | Sample description | Results | Role of LEA | Thematic synthesis |
|---|---|---|---|---|---|---|
| Runyan et al. (2017) | To understand the extent to which LEAs and gun retailers could be viable partners for temporary, voluntary gun storage in communities, particularly when either gun-owning families or health care professionals are concerned about the mental health of household members. | Cross-sectional study | 448 LEAs (sheriff's departments and police agencies) and 95 GUN retailers | Three-quarters of LEAs (74.8%; 95% confidence interval [CI] = 72.1, 77.5) and nearly half of retailers (47.6%; 95% CI = 39.2, 56.0) reported that they currently offered temporary storage of guns. Two-thirds of LEAs (64.6%; 95% CI = 61.5, 67.6) and half of retailers (49.8%; 95% CI = 41.4, 58.2) reported having had requests to provide this service in the past year. Fewer than 40% of either group indicated willingness to store guns without a specific reason for the request. High proportions of both LEAs and retailers were willing to store guns under the scenarios that involved a gun owner concerned about the mental stability of an adolescent in the home (84.8% and 70.5%, respectively) or of an adult family member (83.5% vs. 66.6%), or when the gun owners themselves reported having personal crises (77.7% vs. 67.6%). | Provide lockers for firearm storage | Means Restriction and Environmental Prevention Roles |
| Decker et al. (2018) | To propose a public health framework for preventing community violence, intimate partner violence/ sexual violence and suicide as interrelated forms of violence | NA | NA | Proposes a framework recognising shared risk/protective factors across violence types. Recommends multi-level interventions addressing individual, relationship, community and structural factors. Cross-cutting approaches (e.g., Air Force Suicide Prevention Programme) can address multiple forms of violence. | Police involved in community violence intervention programmes (Cure Violence model) as a suicide prevention strategy: Proactive policing initiatives to address illegal gun possession in high-crime areas at high-risk times reduce gun crimes. Law enforcement is part of survivor support and justice processes and a first responder to crisis (violence and suicide) | Intersectoral Collaboration and Community Engagement, Strategic and System-Embedded Roles of Law Enforcement Agencies |
| Brooks-Russell et al. (2019) | To examine the perceived benefits of and barriers to law enforcement agencies providing increased access to voluntary and temporary firearm storage. | Cross-sectional design | 448 Law enforcement Agencies (306 police departments and 142 Sheriff's Departments) | Although law enforcement agencies perceive structural barriers to providing temporary gun storage, such as space limitations and legal liability, most indicate that suicide prevention is consistent with the mission of law enforcement agencies to promote community safety and are willing to provide gun storage. - Overall, a majority of agencies reported moderate or major benefits to providing storage, including being seen as a positive member of the community, having an opportunity to improve safety and having an opportunity to work with health care. Lack of sufficient space to store guns and potential community distrust associated with law enforcement agencies storing guns. Less commonly reported were logistical, legal, or liability concerns. | Increasing access to voluntary firearm storage by law enforcement agencies, the prevalence of "gun violence restraining orders" to allow police and families to request the removal of firearms in the case of a suicide crisis, understanding and removing barriers to providing firearm storage | Means Restriction and Environmental Prevention Roles |

| References | Objective | Study design | Sample description | Results | Role of LEA | Thematic synthesis |
|---|---|---|---|---|---|---|
| Johnson (2019) | To explore the police officers' experience in dealing with the adolescents who made a non-fatal attempt to commit suicide. | Descriptive Study | 13 police officers (all female) who worked as staff at the suicide prevention helpline/call centre in Guyana | The most frequent concerns presented by adolescent callers to the helpline included: romantic relationship problems, family conflict, sexual orientation conflict and peer pressure. Police Officers reported significant emotional burden when engaging in their counselling role with suicidal adolescents, including experiences of anger, sadness, frustration and despair. | The police officers who operate the suicide hotline are a key group of stakeholders who provide crisis intervention and counselling for suicidal persons; increased training was recommended | Capacity Building and Training-Oriented Engagements, Strategic and System-Embedded Roles of Law Enforcement Agencies |
| Stokoe and Sikveland (2019) | To identify the sequential position, linguistic form and action of the secondary negotiator's interventions on (1) the delivery (e.g. "sound angry") and (2) next actions (e.g. "say please," "try asking them to move") of the primary negotiator and how the primary negotiator incorporates them into the negotiation | Cross-sectional | Police crisis negotiators (primary and secondary), persons in crisis, Metropolitan Police Hostage and Crisis Negotiation Unit, mental health professionals | suggesting alternate or modified first pair parts to encourage PiCs to respond in cases where N1's actions (e.g. questions, summons) did not mobilise a response; suggesting accounts for N1-produced first pair parts (e.g. "say you're finding it difficult to hear"); suggesting answers to PiC questions and suggesting sequence-closing appreciations when PiCs produced preferred responses (e.g. "thank you"). | Police crisis negotiators are the primary actors; primary negotiators communicate directly with persons in crisis; secondary negotiators provide behind-the-scenes support, suggestions and guidance; negotiators must adapt communication strategies in real-time; police negotiation requires specialised training in interactional techniques; successful negotiation depends on how suggestions are formulated and delivered | Capacity Building and Training-Oriented Engagements, Strategic and System-Embedded Roles of Law Enforcement Agencies |
| Osteen et al. (2020) | To examine law enforcement officers' use of suicide prevention strategies and factors impacting their activities | Primary Research (Cross-sectional) | 476 Law Enforcement Personnels | Among 476 law enforcement officers, 64% had prior suicide intervention training, which improved self-efficacy and engagement in suicide risk assessments (70% trained vs. 59% untrained). Despite positive attitudes (90% recognising suicide prevention as part of their role), factual knowledge remained low (54% trained vs. 51% untrained). Nearly 82% of participants reported encountering a suicidal individual as part of their job and approximately 65% of participants felt it was "likely" or "highly likely" they would encounter a person at risk for suicide as part of their job. Scores for self-efficacy to engage in suicide intervention behaviours were high | Incorporating structured suicide prevention training within routine police curricula to strengthen intervention competencies among officers, generating evidence on the advantages of training and self-efficacy on the use of interventions. | Capacity Building and Training-Oriented Engagements |
| Bland et al. (2021) | To examine the impact of the structural reform of the police service and the challenges experienced by officers involved in prevention activity. | Cross-sectional study | 40 interviews with police officers (constable to chief superintendent) and partners (fire, NHS, council, social work, housing, third sector) across 4 local areas in Scotland; 4 focus groups | 14/19 prevention cases were secondary prevention focused on crime/disorder; 4 cases of primary prevention (safety education); only 1 case of tertiary prevention (mental health crisis response); suicide prevention was identified as secondary prevention involving multi-agency learning from past incidents; all prevention required partnerships; collaboration with the fire service and NHS improved outcomes | Police participate in multi-agency suicide reviews to learn lessons and prevent future deaths; work with the fire and the NHS | Intersectoral Collaboration and Community Engagement |

**Table 1.** (*Continued*)

| References | Objective | Study design | Sample description | Results | Role of LEA | Thematic synthesis |
|---|---|---|---|---|---|---|
| Shrestha et al. (2021) | Drawing on the local experience of working in the community on the national level, the study Recommend some best practices on behalf of Transcultural Psychosocial Organisation Nepal to overcome these challenges and to improve the outcomes in our mental health and psychosocial support response to the suicide prevention programme. | | NA | NA | Conducted an information session for police and other stakeholders on suicide and suicide prevention. The role of the police in rescue was much appreciated, where there were no helpline numbers operating. | Capacity Building and Training-Oriented Engagements, Strategic and System-Embedded Roles of Law Enforcement Agencies |
| Ullah et al. (2021) | Aims at analysing the suicide cases that occurred during 2017–2019, uncovering the root causes of suicide and proposing some preventive measures to mitigate the problem. This study highlights gaps in the investigation and prosecution, as well as some social pressures that do not allow the authorities to take suicide cases to their logical conclusions. | Cross-sectional study | 49 suicide cases (2017–2019) from Human Rights Commission Chitral (Secondary data); 16 interview respondents (family members, police, lawyers, clinicians, community members) | Suicide increasing trend (more than doubled in 3 years); 60% female victims; 57% aged 14–26; 55% attributed to mental illness; 25% to domestic violence. Root causes: extended family system, mental health issues, forced marriages, academic pressure and economic factors. Awareness through education and sermons, police complaints centres, the proper prosecution process of suicide cases to convict perpetrators, restructuring extended families into small unit families and imparting lifesaving skills were highlighted as preventive measures to minimise suicides. | Respond to suicidal crises, contribute data on suicide incidents and deaths, supporting suicide surveillance and information systems, maintain public safety and social order in communities, with implications for suicide prevention, work alongside health professionals, social services and other institutional actors in multi-sectoral approaches to suicide prevention | Strategic and System-Embedded Roles of Law Enforcement Agencies, Surveillance, Reporting and Data Systems Role, Intersectoral Collaboration and Community Engagement |
| Hough (2022) | To a brief overview of factors considered in the determination of so-called 'suicide by cop' (SbC), including potential contextual signs of such an event. | NA | NA | NA | Trained to respond to perceived threats, even when SbC is present, but attempt de-escalation. Law enforcement is also positioned as a key partner in post incident- psychological autopsy work (providing information, reports and interviews) | Strategic and System-Embedded Roles of Law Enforcement Agencies |
| Vijayakumar et al. (2022) | to identify key challenges, opportunities and priorities for the national strategy contextualised in the epidemiology and risk and protective factors, to systematically close the gap towards the SDG target for suicide deaths in India | | NA | NA | Police as a source of suicide data through the National Crimes Record Bureau (NCRB) | Surveillance, Reporting and Data Systems Role |

**Table 1.** (*Continued*)

| References | Objective | Study design | Sample description | Results | Role of LEA | Thematic synthesis |
|---|---|---|---|---|---|---|
| Hedman (2023) | To examine interagency collaboration in suicide prevention coordinated by fire and rescue services in Sweden | Ethnographic qualitative study | Fire and rescue service, police, ambulance service, emergency centre, municipality officials | Six central parts identified: (1) shared suicide incident plan, (2) municipality action plan, (3) interagency working groups, (4) proactive MHFA training, (5) emergency response efforts, (6) crisis support for bereaved and first responders. Fire service geographic spread enables faster response. | Police are part of the shared suicide incident plan with the fire and ambulance. Interagency groups include police. Training was conducted for the suicide intervention plan group, including the police. The police create the statistics on suicide in the region | Strategic and System-Embedded Roles of Law Enforcement Agencies, Surveillance, Reporting and Data Systems Role, Intersectoral Collaboration and Community Engagement, Capacity Building and Training-Oriented Engagements |
| Roos and Fjellfeldt (2023) | To map the priorities within Swedish suicide prevention policy at the community level. | | NA | NA | Police identified as key community gatekeepers in need of training | Capacity Building and Training-Oriented Engagements |
| Weiss (2023) | explores the history and evolution of the SbC phenomenon, examines related civil case law and reviews the contours of police-citizen interactions in SbC cases. | NA | NA | NA | Police response to SbC | Strategic and System-Embedded Roles of Law Enforcement Agencies |
| Arya (2024) | To discuss the primary, secondary and tertiary prevention interventions with a focus on the Indian context, while arguing for the need to prioritise primary prevention interventions, at least in the short term, to reduce suicides in India. | NA | NA | Prioritise public health strategies for suicide prevention: enhance surveillance of suicidal behaviour and invest in additional methods such as community and hospital\ based surveillance, focus on restricting access to lethal means such as pesticides, alcohol, hanging points and other materials (in controlled environment), promote responsible reporting of suicide in media by adopting and enforcing WHO guidelines, provide gatekeeper training to relevant individuals such as healthcare workers and education and awareness programmes for various population groups like students | The quality of the NCRB data can be further improved by investment in police resources | Surveillance, Reporting and Data Systems Role |
| Arafat (2025a) | Analyses documents critically, considering initiatives for suicide prevention, highlights the urgent necessity for suicide prevention strategies in the country and identifies prominent stakeholders | NA | NA | NA | Immediate source of suicide data, first responders to suspected suicide deaths | Surveillance, Reporting and Data Systems Role, Strategic and System-Embedded Roles of Law Enforcement Agencies |
| Arafat (2025b) | To discuss the role of the police in suicide prevention in Bangladesh | NA | NA | NA | Declaration of suicide deaths, first responder when a person dies by suicide in a domestic setting; the police should be properly trained | Strategic and System-Embedded Roles of Law Enforcement Agencies, Capacity Building and |

**Table 1.** (*Continued*)

| References | Objective | Study design | Sample description | Results | Role of LEA | Thematic synthesis |
|---|---|---|---|---|---|---|
| | | | | | regarding the risk factors for suicide and the available preventive measures | Training-Oriented Engagements |
| Kennedy (2025) | describes definitions and goals in relation to such incidents and outlines a five-phase framework for their management (training; first responders, preliminary interventions and inquiries; negotiations; resolution; aftercare), indicating the psychiatrist's role during each phase. | NA | NA | NA | Police negotiator in a siege setting (Community Barricaded incident, with a single person threatening suicide; suicide, suicide by cop, suicide-homicide | Strategic and System-Embedded Roles of Law Enforcement Agencies |

**Table 2.** Evidence characteristics of peer-reviewed literature (Intervention-based)

| References | Objective | Study design | Sample description | Intervention description | Mode of Delivery | Results | Role of LEA | Thematic synthesis |
|---|---|---|---|---|---|---|---|---|
| Lee et al. (2015) | To evaluate the joint police–mental health mobile response unit in enhancing the delivery, coordination and quality of community-based crisis response services. | Descriptive mixed-method study | 26 Law enforcement personnel and 11 CAT clinicians | The intervention is a joint mobile crisis response unit in which a police officer and a mental health clinician partner to attend community-based mental health crises together, providing on-site assessment, de-escalation and coordinated decision-making to reduce risks, avoid unnecessary hospitalisations and improve crisis management. | In-person | The 6-month pilot recorded 296 contacts. During this pilot, 33% of the contacts catered to cases of suicide. Police officers reported strong support for the model, noting benefits for consumers, improved support for police and better collaboration across services. | Police and mental health clinicians as a crisis response unit providing on-site assessment, de-escalation, | Intersectoral Collaboration and Community Engagement |
| Arensman et al. (2016) | This study examined the effects of gatekeeper training on attitudes, knowledge and confidence of police officers in dealing with persons at risk of suicide. | prospective single-group pre-test and post-test evaluation | 828 police officers across three European regions of Limerick (Ireland, n = 425 trained out of 643 active-duty officers), Amadora (Portugal, n = 260 trained out of approximately 350 active-duty officers) and Leipzig (Germany, n = 143 out of 1,380 active-duty officers) | Level 1 - Standard Gatekeeper programme Level 2 - Train the Trainer Programme A multi-level suicide prevention approach containing symptoms of depression, warning signs and risk factors associated with suicidal behaviour, motivating help-seeking behaviour, dealing with acute suicide crisis and informing bereaved relatives. | In-person | Results showed significant improvements post-training across all outcomes. Personal depression stigma scores increased from a mean of 32.99 (SD 4.62) to 35.18 (SD 4.81, $F = 100.23$, $p < .001$). Suicide knowledge scores rose from 3.88 (SD 1.27) to 4.35 (SD 1.34, $F = 28.27$, $p < .001$). Confidence to detect suicide risk increased from 4.48 (SD 1.82) to 5.85 (SD 1.80, $F = 248.61$, $p < .001$). Participants with previous training had lower baseline stigma ($M = 32.28$ vs. 33.65), higher knowledge ($M = 4.11$ vs. 3.71) and higher confidence ($M = 4.80$ vs. 4.15) than those without prior training. Four-month follow-up ($n = 46$, 6% response) suggested some decline but scores remained above baseline: DSS 31.57, IKT 3.95, confidence 4.58. | The police force is a key part of the community in terms of their exposure to the most vulnerable and distressed persons in the community. Any intervention that is intended to encourage gatekeeping among community facilitators should include police officers as a target group | Capacity Building and Training-Oriented Engagements |
| Marzano et al. (2016) | The study evaluates a training resource for police officers to assist them in identifying and intervening in situations where an individual may be at risk of suicide. | Mixed-methods study | 168 officers participated in the training programme, | The intervention provides police officers with comprehensive training on suicide prevention, focusing on identifying and managing high-risk individuals, | In-person | Of the 168 trained police officers, 60% completed pre-training and 70% post-training evaluations. Training significantly improved attitudes, knowledge (median score: pre 6/10, post 8/10; $p < 0.0001$) and confidence | Identifying and managing high-risk individuals, understanding risk factors and applying legal and procedural guidelines in varied risk situations, post- | Capacity Building and Training-Oriented Engagements |

**Table 2.** (*Continued*)

| References | Objective | Study design | Sample description | Intervention description | Mode of Delivery | Results | Role of LEA | Thematic synthesis |
|---|---|---|---|---|---|---|---|---|
| | | | | understanding risk factors and applying legal and procedural guidelines in varied risk situations. It includes modules on approaching individuals at risk, assessing risk levels and referring them to suitable services. Additionally, it covers post-incident procedures, such as supporting bereaved families, handling media and providing mental health resources for officers. This training enhances officers' skills in suicide prevention and response within their specific role and context. | | across all suicide prevention domains, with positive changes sustained at 6-month follow-up. | incident procedures, such as supporting bereaved families, handling media | |
| (Gilissen et al. 2017) | The study describes the need to use evidence-based suicide prevention approaches on multiple levels and illustrates the multilevel community approach to prevent suicides | cross-sectional survey and programme evaluation | 7 regions in the Netherlands. | Component 1: The public awareness campaign on talking about suicide, recognising the signs of suicidality and managing suicidality. Component 2: Training local gatekeepers to identify suicidal risk, to make contact and to "open the gates" for help-seeking behaviour. Component 3: Targeting high-risk groups Component 4: Support of professionals in general practice to explore suicidal feelings during consultations with patients with depression. | In-person | The study found that the SUPRANET Community programme increased awareness of the Dutch helpline 113 Suicide Prevention and reduced depression stigma in intervention regions compared to the control region. Although differences between high- and medium-exposure regions were not statistically significant, the programme effectively promoted attitudinal change, which is crucial for encouraging help-seeking among those with suicidality. The study found that primary care professionals (PCPs) valued the SUPRANET Community training for improving their knowledge and communication skills in suicide prevention. However, PCPs faced challenges in | Community facilitators, such as police officers, throughout the country have been trained as gatekeepers | Capacity Building and Training-Oriented Engagements |

*Cambridge Prisms: Global Mental Health*

| References | Objective | Study design | Sample description | Intervention description | Mode of Delivery | Results | Role of LEA | Thematic synthesis |
|---|---|---|---|---|---|---|---|---|
| | | | | | | identifying suicide risk and expressed a need for better collaboration with mental health services to effectively implement suicide prevention practices. | | |
| Bouveng et al. (2017) | To describe the first-year activity and outcomes of the Psychiatric Emergency Response Team (PAM) providing prehospital psychiatric care in Stockholm | Retrospective cross-sectional mixed | NA | specialised prehospital mental health response unit integrated into the Emergency Medical Service (EMS) system in Stockholm County. It operates as an alternative to traditional police and ambulance responses for psychiatric and mental health emergencies. | In-person | The Police Department in Stockholm County registered a total of 3,271 emergency cases regarding suicide (communication/attempt) and 1,128 cases regarding acute mental illness/distress during the first year of PAM. Out of 1,580 requests to PAM, 80% resulted in attended cases and 97% of all requests had high or medium priority (Priority level 1 or 2). The PAM team responded to psychiatric emergencies, including suicidal ideation, severe mental illness and substance abuse. The team provided prehospital assessment and reduced unnecessary emergency department visits. The most common causes were suicidal ideation and mental illness. Cooperation with other departments was common, mainly ambulance (55%) and police (49%), but rescue services were also involved in some cases (7%). Only 24% of the cases were handled without other services involved. | Collaborative partners on psychiatric emergency calls; focus on scene safety, backup for high-risk situations and transport when needed; recommendations to reduce police-only psychiatric responses and integrate within specialised mental-health-led models | Intersectoral Collaboration and Community Engagement, Strategic and System-Embedded Roles of Law Enforcement Agencies |
| McGeechan et al. (2017) | The objective of the study is to determine whether a pilot police-led suspected suicide surveillance system improves the speed and number of reported suspected suicides compared with the existing coroner-based surveillance system and whether it leads to an | Mixed-methods study | Bereaved individuals | The police-led pilot suicide surveillance strategy follows a similar process to the coroner-led strategy in terms of completion of the notification of death form. However, in cases where the officer feels the death may be a suspected suicide, they will discuss postvention | In-person | Most suspected suicides were captured by coroner reports (94.2%) and a smaller proportion by police NoD forms (78.8%). Reporting was substantially faster through the police system, enabling earlier alerts to support services. Among those reported through NoD forms, the majority (78%) consented to share details, which led to | A police-led real-time suspected suicide surveillance strategy has the potential to provide a real benefit to those bereaved by suicide by providing quicker referral, leading to faster support | Surveillance, Reporting and Data Systems Role |

**Table 2.** (*Continued*)

| References | Objective | Study design | Sample description | Intervention description | Mode of Delivery | Results | Role of LEA | Thematic synthesis |
|---|---|---|---|---|---|---|---|---|
| | increase in referrals to support services following a suspected suicide. | | | support with those affected by the death and request consent to record their contact details on an additional line on the notification of death form. The police officer then sends the notification of death form to the coroner's office and NECS as per the coroner-led strategy. Where consent is given for referral to support services, the notification of death form is also shared with the local authority's Public Health suicide prevention lead, who makes a referral to support services. Bereaved individuals are then contacted within 2 days by a postvention support service and offered bereavement support, which may include therapeutic support, financial advice and legal advice. | | referrals that supported multiple affected individuals. Referrals during the pilot increased by 88% compared to the previous year, with roughly three-quarters originating from NoD forms. All referred individuals accessed bereavement support services, with smaller proportions using welfare rights, advocacy, additional bereavement support and legal assistance. Qualitative Findings Police officers reported limited clarity and confidence in explaining the pilot and some discomfort seeking consent at the scene, which hindered consistent use of NoD forms. In contrast, support services described the pilot as enabling much earlier engagement and substantially increased referrals. They suggested integrating the NoD form into standard sudden-death packs to improve uptake. Multiagency collaboration was highlighted as a key strength, enhancing coordination and the quality of postvention support. | | | |
| Jager-Hyman et al. (2019) | To explore the perspectives of firearm stakeholders on promoting secure firearm storage in paediatric primary care as a universal suicide prevention strategy | | | The Firearm Safety Check is an evidence-based violence prevention intervention designed to promote safe firearm storage practices, including: (1) screening for the presence of firearms in the home; (2) brief motivational interviewing informed counselling regarding safe storage; and (3) provision of free cable locks. | In-PERSON | Stakeholders identified barriers (political polarisation, liability concerns) and facilitators (framing as safety, universal approach, not requiring disclosure) for firearm safety promotion. Recommendations for adapting interventions to increase acceptability among firearm owners. | Credible partner and messenger for safe-firearm-storage promotion; build trust between firearm-owning communities and health systems; provide reassurance about privacy and non-enforcement intent of clinical interventions | Means Restriction and Environmental Prevention Roles, Intersectoral Collaboration and Community Engagement |

(*Continued*)

| References | Objective | Study design | Sample description | Intervention description | Mode of Delivery | Results | Role of LEA | Thematic synthesis |
|---|---|---|---|---|---|---|---|---|
| Ross et al. (2020) | This study aimed to evaluate the effectiveness of the Gap Park Masterplan in reducing suicides through the application of a convergent parallel mixed-methods design | Mixed-methods study | Residents of Sydney | The intervention involved the implementation of a series of suicide prevention initiatives, including fencing, CCTV surveillance, protocols with police and the installation of phone booths and promotion of the Lifeline Suicide Hot Spot Emergency phone service. | In-person | The findings of the study showed a non-significant upward trend in jumping suicides overall during the study period. For females, there was a significant upward trend in suicides between 2000 and 2010 (APC = 16.64%,95% CI:8.18, 25.76, $p < 0.001$), followed by a significant downward trend (APC = $-$21.27%,95% CI: $-$33.14, $-$7.30, $p = 0.01$) after its implementation (2010–2016). However, a non-significant upward trend for males was observed (APC = 6.23%,95%CI: $-$0.41,13.30, $p = 0.06$). Qualitative interviews with police officers revealed six key themes, including romanticism, behavioural patterns of suicidal individuals, means restriction and personal impacts on officers. CCTV and alarms were found to be effective for detecting and locating suicide attempts. | Suicide prevention initiatives, including fencing, CCTV surveillance and protocols with police | Means Restriction and Environmental Prevention Roles, Surveillance, Reporting and Data Systems Role, Strategic and System-Embedded Roles of Law Enforcement Agencies |
| Blais and Brisebois (2021) | The study evaluates the effect of a co-response police-mental health programme introduced by the Laval Police Department to improve interventions in suicide-related calls. | RCT | 800+ Police officers | The study used a two-pronged service: At level 1 - Police officers were trained on social vulnerabilities and suicide related crisis. Training addressed legal issues regarding interventions with people in crisis, importance to undertake the suicide risk assessment. The training also raised awareness about negative consequences that can result from failure to contact the police services when confronted with a suicide-related behaviour. At level 2, | In-person | Results indicate that the co-response programme was associated with significant decreases in police use of force (ATE = $-0.077$; $p \leq 0.05$) and transports to hospital (ATE = $-0.773$; $p \leq .01$). Increases were observed in referrals to community resources (ATE = 0.285; $p \leq 0.01$) and individuals managed through their social network (ATE = 0.530; $p \leq 0.01$). | Police officers were trained on social vulnerabilities and suicide related crisis. | Capacity Building and Training-Oriented Engagements |

**Table 2.** (*Continued*)

| References | Objective | Study design | Sample description | Intervention description | Mode of Delivery | Results | Role of LEA | Thematic synthesis |
|---|---|---|---|---|---|---|---|---|
| | | | | Psychosocial workers receive training on best practices for suicide-related interventions. Training gives the knowledge and ability to identify individuals at risk of suicide, create therapeutic relationships, explore criteria to estimate suicide risk, work on ambivalence and repositioning, facilitate access to services and offer immediate and short-term follow-up. Training incorporates role-play, case studies and group discussions. " | | | | |
| Hofmann et al. (2021) | To evaluate the efficacy of a suicide prevention online training for police officers. | quasi-experimental study | 102 Law enforcement personnel | The intervention was a three-module online training for police officers, covering death notification, responding to individuals with suicidal ideation and managing personal stress and suicidal thoughts, delivered as self-paced psychoeducational content with practical guidelines, case examples, worksheets and quizzes. | Online | The training led to significant gains in overall self-rated competence (from $M$ = 68.63 to 79.79, $p$ < 0.001) and knowledge (from $M$ = 22.44 to 26.49, $p$ < 0.001), with all competence and knowledge subscales improving similarly. Attitudes towards suicide and mental health symptoms showed no significant change (all $p$ > 0.05). Moderation analyses found no effects of years in service or work location and participants rated the programme highly (88–93% reporting it was helpful or well-prepared), despite substantial attrition (102/194 completing post-training) | Role in death notification, responding to individuals with suicidal ideation | Strategic and System-Embedded Roles of Law Enforcement Agencies |
| Ko et al. (2021) | To investigate the effect of suicide prevention education on attitudes towards suicide among police officers | Cross-sectional survey design | 518 police officers | The intervention is an education programme titled "Observing, Listening and Speaking." It teaches core gatekeeper skills by breaking down each component of the title. "Observing" focuses on early identification of | In-person | 247 out of 518 officers (47.7%) had received suicide-prevention training. Trained officers were significantly more likely to view suicide as predictable, disagreeing more with the ATTS item "suicide is unpredictable" (3.36 vs. 3.35; $p$ = 0.001). They were also more likely to see suicide as a cry for | Police, as frontline officers who work in duty related to individuals at risk of suicide and suicide prevention education is associated with positive attitudes towards suicide and suicide prevention in the police | Capacity Building and Training-Oriented Engagements |

| References | Objective | Study design | Sample description | Intervention description | Mode of Delivery | Results | Role of LEA | Thematic synthesis |
|---|---|---|---|---|---|---|---|---|
| | | | | warning signs. "Listening" trains participants in active, empathetic engagement with individuals at risk. "Speaking" introduces safety checklists, referral pathways and collaboration with mental health professionals. | | help and to reject the belief that "suicidal thoughts never disappear" (2.08 vs. 2.26; p = 0.025; 3.20 vs. 3.05; *p* = 0.035). | | |
| Dawson et al. (2021) | To determine the feasibility of conducting data linkage across key criminal justice datasets (police, prison, probation) to establish a national suicide database and outline the processes, methodological considerations and any other implications of setting up such a linkage. | | Police National Computer (PNC) • Perito; system for Independent Office for Police Conduct (IOPC) • Offender Assessment System (OASys) • Prison National Offender Management Information System (P-NOMIS) • nDelius (national probation case management system) | Linkage of the ONS suicide data with the following five UK criminal justice datasets: • Police National Computer (PNC) • Perito; system for Independent Office for Police Conduct (IOPC) • Offender Assessment System (OASys) • Prison National Offender Management Information System (P-NOMIS) • nDelius (national probation case management system) | NA | All five criminal justice datasets were deemed feasible for linkage. Discovery of the 'spine' dataset enables effective record matching across police, prison and probation identifiers. The proposed linkage could identify key intervention timepoints for suicide prevention. | Core data provider through Police National Computer; custody-related data and death investigation records; recommendations to improve data quality, standardise risk-flag recording and link with prison/probation/mortality data to identify intervention points | Surveillance, Reporting and Data Systems Role |
| Osteen et al. (2021) | The purpose of this study was to deliver an online QPR training to improve knowledge about suicide, attitudes to suicide and suicide intervention and self-efficacy. Additionally, the study assesses the association of these outcomes with greater use of intervention skills when encountering individuals at risk for suicide in the community. | Quantitative | 108 Police Personnel | The study utilised Question, Persuade and Refer (QPR) for Law Enforcement. The base training focused on detecting and referring someone at risk of suicide for appropriate care. It equipped LEA to recognise and respond positively to someone exhibiting suicide warning signs. The advanced training expands LEA skills in suicide risk assessment and management. | Online | The study found a statistically significant increase in suicide intervention knowledge among law enforcement officers after QPR training (mean 5.10 to 5.79, $F = 16.04$, $p < 0.001$) and a significant reduction in negative attitudes (mean 25.36 to 22.14, $F = 11.68$, $p = 0.002$), both after controlling for prior training. Self-efficacy did not change significantly (mean 5.46 to 5.33, $F = 2.48$, $p = 0.11$), nor did the frequency of suicide intervention behaviours, which were already high at baseline and remained stable at three months. | LEO plays an important role in suicide intervention and thus need appropriate training to engage with these at-risk individuals | Capacity Building and Training-Oriented Engagements |

| References | Objective | Study design | Sample description | Intervention description | Mode of Delivery | Results | Role of LEA | Thematic synthesis |
|---|---|---|---|---|---|---|---|---|
| Hill et al. (2022) | To examine the reach and perceived effectiveness of the Primary Care Navigator (PCN) active outreach postvention model for people bereaved by suicide | Retrospective cross-sectional mixed methods | 80 suspected suicides; 347 bereaved individuals; interviews with 5 bereaved, 18 stakeholders, 5 WA police officers (Park region, Australia) | The Primary Care Navigator (PCN) model is an active outreach postvention intervention that provides acute support to people bereaved by suicide in the immediate aftermath of a death (within 48–72 h). Rather than requiring bereaved individuals to initiate contact with services themselves (passive postvention), the PCN proactively reaches out and offers support (active postvention) (initiated by police) | In-Person | 80 suspected suicides; 347 bereaved received outreaches; 164 (47.2%) accepted further support. Support included: bereavement information (98%), clinical support (49.6%), postvention counselling (38.4%) and financial assistance (16%). Police, stakeholders and bereaved perceived the PCN model as effective. | Key initiator of Primary Care Navigator postvention model; identify bereaved individuals and trigger active outreach; provide data for suicide cluster monitoring; core partner in coordinated postvention response | Strategic and System-Embedded Roles of Law Enforcement Agencies, Surveillance, Reporting and Data Systems Role, Intersectoral Collaboration and Community Engagement |
| (Wong et al. 2022) | The objective of this chapter is to demonstrate how the Hong Kong Police Force's Negotiation Cadre applies the revised Behavioural Influence Stairway Model (BISM–2) in real-world suicide crisis negotiations, using a recent case of preventing a jump attempt from a residential building. | | The Police Negotiation Cadre of the Hong Kong Police Force | The Hong Kong Police Force's Police Negotiation Cadre (PNC) is a 24/7 crisis-response team composed of carefully selected officers who undergo a rigorous one-day selection process, psychological screening and a two-week (120-h) intensive training programme focused on crisis and suicide negotiation. Their training integrates the BISM–2 framework and core competencies such as active listening, empathy, rapport-building, trust development and strategic influence techniques aimed at facilitating behavioural change during high-risk interactions. Newly trained officers complete supervised callouts and receive continuous advanced | In-person | The negotiation team successfully applied BISM–2 using active listening, empathy, rapport, trust-building and influence to de-escalate the crisis and prevent the individual from jumping, ultimately achieving a safe resolution. The case also demonstrates the practical utility of structured negotiation models in real-world, high-risk police interventions. | crisis and suicide negotiation. | Strategic and System-Embedded Roles of Law Enforcement Agencies |

*Cambridge Prisms: Global Mental Health*

| References | Objective | Study design | Sample description | Intervention description | Mode of Delivery | Results | Role of LEA | Thematic synthesis |
|---|---|---|---|---|---|---|---|---|
| | | | | training from international experts to maintain and strengthen their skills. | | | | |
| Yeung et al. (2022) | To document and evaluate the short-term and long-term effects of a multidisciplinary, multilayer, community-based suicide prevention programme implemented in the Eastern District of Hong Kong and assess programme sustainability. | | Eastern District of Hong Kong (intervention area) population during 2008–2012 intervention period and 4 years post-intervention (2012–2016); comparison with the remainder of the Hong Kong population | This multi-level community-based suicide prevention model had three operational levels: indicated (targeting individuals at risk), selective (targeting at-risk subgroups) and universal (targeting the whole population) and five elements: 1. support people bereaved by suicide; 2. target individuals with self-harm behaviours; 3. train gatekeepers (healthcare professionals, police officers, social workers, etc.); 4. raise community awareness; and 5. Establish a referral system to social services | In-person | (1) Short-term effects (during intervention 2008–2012): Suicide rates in Eastern District were significantly lower compared to the rest of Hong Kong; self-harm rates continuously dropped and remained lower; pattern shifts observed in method of suicide (more deaths from jumping, fewer by charcoal burning); age of people dying by suicide increased. (2) Long-term effects (post-intervention 2012–2016): Rates slowly rebounded after intervention ceased but remained lower than the rest of Hong Kong through 2016; self-harm rates continued to drop and remained persistently lower than the rest of Hong Kong; sustainability was limited without continued intervention components. (3) Differential impacts by age and gender: Programme impacts varied significantly by age and gender subgroups, indicating that different population segments responded differently to interventions. (5) Spatial analysis: Geographic areas where interventions were concentrated showed persistent suicide rate reductions, suggesting location-specific effects. | Data contribution on suicide/attempts; participate in multi-agency coordination; serve as gatekeepers identifying and referring at-risk individuals; training and collaboration with health and social sectors | Capacity Building and Training-Oriented Engagements, Surveillance, Reporting and Data Systems Role, Intersectoral Collaboration and Community Engagement |
| Thorne and O'Reilly (2022) | To investigate a model based on real-time surveillance collection of data, for suicide prevention, intervention and postvention. | Multiple-case study design | the police, mental health services, local authorities, charities and academics in England. Two UK police forces (multiple case-study sites); participants include police personnel and | The LOSST LIFFE (Learning from Officers' Suicide Support Tasks: Leicester Investigation of a Framework for Family Engagement) model is based on real-time surveillance collection of data for | In-PERSON | (1) Current data collected on deaths by suicide is limited in scope; (2) More localised responses to suicide prevention are necessary; (3) Multi-agency communication is essential for effective implementation; (4) Utilisation of existing local support | Collection of real-time surveillance data related to suicide by the police to facilitate quicker future response by legal and mental health professionals, provision of suicide awareness training to | Surveillance, Reporting and Data Systems Role |

**Table 2.** (*Continued*)

| References | Objective | Study design | Sample description | Intervention description | Mode of Delivery | Results | Role of LEA | Thematic synthesis |
|---|---|---|---|---|---|---|---|---|
| | | | staff from real-time surveillance and suicide prevention units | prevention, intervention and postvention based on a systemic community-focused operational approach. The implementation of this model provides a mechanism for acquiring demographic details, characteristics of the deceased and location to create a database that can be used for mobilising action for crisis management, intervention and management. and communication strategies for providing individual support and facilitating connections with other local agencies for tailored and relevant support at a community and public service level. | | systems improves outcomes; (5) Emotional support for frontline practitioners is critical; (6) Police uniquely positioned on the frontline to collect data and support families | relevant staff by police, engagement of police in multi-agency communication with other services and sectors, including NHS (National Health Service), provision of bereavement support by partnering with health services and charities | |
| Young et al. (2022) | to employ a scenario-based training, conclude with a group debriefing of the scenario and educate about responding to armed suicidal subjects. The current study also sought to quantify typical police officer responses to a suicidal subject armed with a pistol, to examine the impact of de-escalation training on officer intervention with suicidal subjects and to quantify police officer safety issues that might arise when officers respond to an armed suicidal subject. | Quasi-experimental study | 108 officers and deputies | Each officer was given a pistol loaded with simulated munitions (a pistol that fired plastic projectiles that marked where they hit with a blue powder) and was then led to an open field with a single role player facing away from them Before entering the scenario and standing more than 45 ft. away, the participants were given the same briefing: You are responding to a suicidal subject who is a veteran possibly armed with a pistol. You have no cover (a place to hide behind that would stop bullets), no | In-Person | The suicidal subject scenario was stopped by the instructors 76.2% of the time, ended with the officer removing the gun from the suicidal subject's hand 32.1% of the time and ended with an officer shooting the suicidal subject 27.8% of the time. Seventy-eight percent of participants drew their weapon when they could see a pistol in the role player's hand and 86.2% closed the distance, or allowed the role player to close the distance between them during the scenario. Almost 58% of participants ended up walking backwards or in a circle as the role player closed the distance between them. Just over 8% of | Police are first responders to armed suicidal subjects; officers must assess lethality and make rapid decisions; crisis negotiators are specialised LEA trained in de-escalation; police must balance public safety with subject welfare; need for specialised training in recognising and responding to suicide by cop scenarios | Capacity Building and Training-Oriented Engagements |

*Cambridge Prisms: Global Mental Health*

| References | Objective | Study design | Sample description | Intervention description | Mode of Delivery | Results | Role of LEA | Thematic synthesis |
|---|---|---|---|---|---|---|---|---|
| | | | | concealment and no backup officers coming. The participant was then advised of the boundaries of the open field and that if they reached that boundary, the researcher would put their hand on them and advise them to move in a different direction. Two of the boundaries were an "open roadway" and a "residential area." The participant was asked to respond to this suicidal subject call as they would any patrol call for service and then asked if they had any questions. Once the participant was ready, they were walked to the 45-ft starting position and the scenario began. | | participants never said anything to the role player. | | |
| Marzano et al. (2023). | To set out the nature, status and content of a real-time suicide monitoring system in Great Britain (England, Scotland and Wales) and explore its potential to contribute to timely and targeted suicide prevention initiatives. | | National Police Chiefs' Council (NPCC), British Transport Police (BTP) | The system collects standardised data on all suspected suicides reported by police forces using a centralised secure email system to a central BTP team. Data includes demographic information (age, gender, nationality), life events, mental health problems, previous police contact, circumstances of death (time, date, location, method) and location of death versus home address. Reports are compiled monthly with a one-month lag, creating the most timely and comprehensive overview of suicides in | In-PERSON | (1) By December 2022, police-led surveillance system achieved 98% population coverage of Great Britain (England, Scotland, Wales); (2) System provides most timely and comprehensive suicide overview currently available in UK; (3) Data compiled with one-month time-lag (fastest available); (4) Records demographic information (age, gender, nationality), life events, mental health problems, previous police contact; (5) Records both location of death and deceased's home address (vs. official statistics using only residence); (6) Identified high-risk locations/hotspots: one local authority area showed suicide rate > double the next highest area; (7) Data completeness: ~98% for main | Police should continue to serve as primary data collectors given their legal obligation to attend all sudden, unexpected deaths and existing infrastructure. | Surveillance, Reporting and Data Systems Role |

**Table 2.** (*Continued*)

| References | Objective | Study design | Sample description | Intervention description | Mode of Delivery | Results | Role of LEA | Thematic synthesis |
|---|---|---|---|---|---|---|---|---|
| | | | | the UK. The system records deaths by date of occurrence rather than registration date, enabling rapid identification of trends, clusters and emerging patterns. Data quality is maintained through standardised assessment guidance based on 'Overstone' criteria for determining suspected suicide classification. | | fields (age, gender, method, circumstances of death); (8) System enables identification of suicide clusters and imitation effects; | | |
| Olibamoyo (2023) | The objective of the study is to assess changes in police constables' knowledge of suicide, self-rated confidence in enquiring about suicidal behaviours and attitudes towards suicidal behaviours following brief suicide intervention training. | quasi-experimental study | 289 police constables | The study implemented a suicide prevention training, including didactic components focusing on developing communication skills with potentially at-risk people. The didactic component consisted of education on suicide and other psychological symptoms prevalence, risk factors, warning signs and myths. Training provided participants with education on how to ask questions about suicide and other psychological symptoms and refer people to resources, including specific decision trees and scripts that could be used. | in-person | Police constables showed a significant increase in perceived knowledge about suicide (QPR mean 22 → 26.4, $p < 0.001$), self-confidence in asking about suicidal ideation (CBQ mean 11 → 12, $p < 0.001$) and attitudes towards suicidal behaviours (ATTS mean 10 → 13, $p < 0.001$) following training, while actual knowledge (SIT mean 5.5 → 5.5, $p = 0.8$) did not change. | Police, as key stakeholders in suicide prevention, are trained in QPR | Capacity Building and Training-Oriented Engagements |

**Table 3.** Evidence characteristics of grey literature consisting of policy and guidance documents

| References | Geographical Scope | Scope of document/objective | Key Roles and Responsibilities of LEAs | Thematic Synthesis |
|---|---|---|---|---|
| World Health (2009) | International Guidance | The booklet places suicide in the broader context of community mental health and identifies several principles and key activities that can be used as part of a broader community-based suicide prevention strategy by police, firefighters and other first responders | Knowledge about risk factors of suicide and related laws as a first responder, restricting access to lethal means, risk assessment and ensuring firearm safety, referrals to appropriate mental health services, admission to hospital under mental health legislation, management and de-escalation of suicide by cop, contacting family and friends of the deceased and providing adequate support or referrals after crisis. Also provides post-attempt support by dealing with the crisis, providing basic help and arranging for the person to be transferred to a health centre | Capacity Building and Training-Oriented Engagements, Intersectoral Collaboration and Community Engagement, Strategic and System-Embedded Roles of Law Enforcement Agencies, Means Restriction and Environmental Prevention Roles |
| (World Health, 2010) | International Guidance | To outline fundamental principles for developing and evaluating community suicide prevention programmes | Gatekeeper training for police | Capacity Building and Training-Oriented Engagements |
| (World Health, 2012) | International Guidance | a resource to assist governments to develop and implement a strategy for suicide prevention as well as to help those that have already begun the process of conceptualising national suicide prevention strategies | Police identified as a first responder, a comprehensive training programme for identified gatekeepers, including police, a multi-sectoral approach, including multiple sources such as police, to gather suicide data. | Capacity Building and Training-Oriented Engagements, Surveillance, Reporting and Data Systems Role, Intersectoral Collaboration and Community Engagement |
| (Ministry of Health Malaysia, 2013) | National Policy | It is a guideline for suicide prevention and management to be used by agencies and institutions such as healthcare facilities, prisons, schools. It aims to promote a collaborative approach in the prevention of suicide for frontliners, increase suicide awareness, introduce the use of Suicide First Aid and provide a guide for management, training, monitoring and research in suicide. | Police as an emergency response service to suicide, intersectoral collaboration with the suicide acute response team | Strategic and System-Embedded Roles of Law Enforcement Agencies, Intersectoral Collaboration and Community Engagement |
| (Suicide Prevention Resource Center, 2013) | National Guidance | Comprehensively addresses law enforcement roles across the full continuum of suicide prevention | crisis intervention skills for rapport-building and risk assessment, documentation of findings at the site of crisis, postvention support to survivors of suicide, interacting with and providing support to family and friends of deceased or suicidal individuals | |
| (Australian Government Department of Health, 2014) | National Policy | To evaluate self-reported achievement of National Suicide Prevention Programme (NSPP) funded projects across six LIFE Action Areas. Assessment examined how effectively projects targeted their suicide prevention work across six key areas. Results showed projects addressed multiple action areas with average achievement scores ranging from 3.41 to 4.06. | Police identified as first responders requiring gatekeeper training. Intersectoral-collaboration between police and mental health services. Police are critical for identifying individuals at risk and making referrals to mental health and support services; engage in crisis response and emergencies involving individuals expressing suicidal ideation; contribute data on suicide and self-harm incidents to surveillance and monitoring systems; engage in community-based prevention activities and coordinated local responses to suicide hotspots and prevention priorities. | Capacity Building and Training-Oriented Engagements, Surveillance, Reporting and Data Systems Role, Intersectoral Collaboration and Community Engagement |
| (National Human Rights Commission India, 2014) | National Guidance | Examines suicide in custody across India, with a specific focus on relevant professionals and staff to work towards suicide prevention in prisons and cells | Intake screening and assessment of arrestee/inmate, communication between the arresting LEA and prison staff, close and constant supervision of arrestee/inmates depending on the level of risk, ensuring a safe environment inside the cell, completing referrals to appropriate mental health services. | Intersectoral Collaboration and Community Engagement, Means Restriction and Environmental Prevention Roles, Strategic and System-Embedded Roles of Law Enforcement Agencies |

(*Continued*)

**Table 3.** (*Continued*)

| References | Geographical Scope | Scope of document/objective | Key Roles and Responsibilities of LEAs | Thematic Synthesis |
|---|---|---|---|---|
| (Ministry of Health and Social Services, 2015) | National Policy | This strategy document aims to reduce suicide and self-harm in Wales. It sets out the strategic aims and objectives to prevent and reduce suicide and self-harm in Wales over the period 2015–2020. It identifies priority care providers to deliver action in certain priority places to the benefit of key priority people and confirms the national and local action required. | Bereavement support and support for people who attempted suicide | Strategic and System-Embedded Roles of Law Enforcement Agencies |
| (Ministry of Health Canada, 2016) | National Policy | This report provides an update on suicide prevention initiatives across the federal departments and agencies and Communities of Canada. It reports activities that aim to promote mental health, reduce stigma and raise public awareness, accelerate research and innovation in suicide prevention | Training (Suicide awareness and prevention training) for officers who are likely to encounter people living with serious mental stress or considering suicide (first responders) | Capacity Building and Training-Oriented Engagements |
| (The Columbia Lighthouse Project, 2016) | National Guidance | Positions suicide risk assessment as a critical, learnable skill for first responders (police, firefighters, EMS/paramedics) to enable early identification of at-risk individuals, determine appropriate response level and prevent suicide deaths. | Use of the Columbia-Severity Rating Scale as a risk detection tool by frontline workers, including police officers. It provides a common language for understanding the level of risk, enabling the linkage of systems, which facilitates care delivery. It also provides the two- to six-question screener, which makes it easier for first responders to ask questions without any prior mental health training. | Intersectoral Collaboration and Community Engagement |
| (Royal Government of Bhutan, 2018) | National Policy | addresses relevant stakeholders to strengthen and scale up suicide prevention through a coordinated national programme that emphasises multi-disciplinary actions: upskilling frontline workers, school- and community-based prevention, helplines, surveillance enhancements and reducing access to means. | Suicide Prevention Unit of the Royal Bhutan Police (RBP) for coordination of suicide prevention programmes across Bhutan, detecting psychological distress and referral to appropriate support services and crisis intervention during suicidal crises. Provide data for the national suicide registry | Intersectoral Collaboration and Community Engagement, Surveillance, Reporting and Data Systems Role, Strategic and System-Embedded Roles of Law Enforcement Agencies |
| (Department of Health, 2019) | National Policy | A long-term strategy for reducing suicides and the incidence of self-harm with action delivered across a range of Government departments, agencies and sectors. | Suicide prevention training for police custody staff, police record basic information about the deceased in the Sudden Death Notification form, which is then shared with the local Health Trust and the Public Health Agency. Police may provide immediate bereavement and postvention support. Police hold intelligence (Data) that is relevant to understanding the context and patterns of suicide | Strategic and System-Embedded Roles of Law Enforcement Agencies, Capacity Building and Training-Oriented Engagements, Surveillance, Reporting and Data Systems Role, Intersectoral Collaboration and Community Engagement |
| (Illinois Department of Public Health, 2019) | State Policy | Highlights the crucial role of first responders (police, fire, EMS) in recognising, intervening and referring individuals at risk of suicide, as well as supporting the well-being of responders themselves. | Recognising warning signs of suicide, refer suicidal individuals to a mental health facility, remove lethal means to ensure safety, get involved in local prevention efforts at the community level, and promote awareness | Intersectoral Collaboration and Community Engagement, Means Restriction and Environmental Prevention Roles, Strategic and System-Embedded Roles of Law Enforcement Agencies |
| (National Suicide Prevention Alliance, 2019) | National Policy | This practice resource is to support local authority public health teams to work with sustainability and transformation partnerships (STPs) and integrated care systems (ICSs), health and wellbeing boards, the voluntary sector and wider networks of partners to implement local suicide prevention plans and embed | Multi-agency partnership for suicide prevention, including police. The police's role in identifying factors that influence suicide risk | Intersectoral Collaboration and Community Engagement |

(*Continued*)

| References | Geographical Scope | Scope of document/objective | Key Roles and Responsibilities of LEAs | Thematic Synthesis |
|---|---|---|---|---|
| | | work within local sustainability and transformation plans. | | |
| (Police Executive Research Forum, 2019) | National Guidance | A tool for police officers to recognise and respond safely to incidents of "Suicide by Cop" (SbC). | Response to a Suicide by Cop (SbC) crisis call effectively, ensuring the safety of everybody present at the site of the crisis, providing critical information to responding officers regarding the SbC crisis and effective communication with suicidal individuals | Strategic and System-Embedded Roles of Law Enforcement Agencies |
| (The Mental Health Services Oversight and Accountability Commission, 2019) | State Policy | This is a comprehensive strategy that incorporates the latest information and evidence to guide state and local actions for suicide prevention in California. Its objective is to equip and empower California communities with the information they need to minimise risk, improve access to care and prevent suicidal behaviours | Part of death review teams for clinical and forensic review of suicide deaths, guide dissemination of lawful options for temporarily transferring firearms for storage in times of suicide crisis or when Gun Violence Restraining Orders apply, training in best practices for messaging following a suicide, provide more detailed information on the circumstances surrounding suicide, suicide prevention in crisis intervention training, as well as co-responder models and supporting people bereaved by suicide | Strategic and System-Embedded Roles of Law Enforcement Agencies, Intersectoral Collaboration and Community Engagement, Means Restriction and Environmental Prevention Roles, Capacity Building and Training-Oriented Engagements |
| (Shaw, 2021) | National Guidance | Applies an intersectional and racial equity lens to critically assess law enforcement's role in suicide prevention programmes and practices. | crisis intervention training for police officers as part of a crisis response team with clear criteria for involvement of law enforcement: creating mobile crisis response teams involving police, school tip-lines with mechanisms for directing tips involving suicide risk and mental health to qualified professionals, expanding and elaborating on Extreme Risk Protection Orders (ERPOs) petition and surrender processes. | Capacity Building and Training-Oriented Engagements, Intersectoral Collaboration and Community Engagement |
| (Bertrand, 2022) | National Guidance | addresses suicide prevention competencies for law enforcement personnel (and all first responders) through two interconnected evidence-based gatekeeper training programmes | Having open conversations about suicide, identifying signs and crisis intervention by creating a safety network of professionals trained in suicide prevention | Capacity Building and Training-Oriented Engagements |
| (Ministry of Health and Family Welfare, 2022) | National Policy | India's National Suicide Prevention Strategy aims to reduce the suicide mortality by 10% in the country by 2030. It is envisioned to do this by reinforcing leadership and institutional capacity, enhancing the capacity of health services, developing community resilience and societal support and strengthening surveillance and evidence generation. | Training community stakeholders, like police, to strengthen health systems for suicide prevention | Capacity Building and Training-Oriented Engagements |
| (Scottish Government, 2022) | National Policy | This strategy sets out the Scottish Government and COSLA's vision for suicide prevention in Scotland over the next ten years; to reduce the number of suicide deaths in Scotland, whilst tackling the inequalities which contribute to suicide. | Police are recognised as a first responder to suicide who will be a part of a Delivery Collective, which brings together delivery partners across Scotland to learn, connect and take a joined-up strategic approach to deliver the national actions; real-time suicide and self-harm data will be provided by the police | Surveillance, Reporting and Data Systems Role Intersectoral Collaboration and Community Engagement |
| (Department of Health and Social Care, 2023) | National Policy | This cross-government strategy aims to bring national government, NHS, local government, the voluntary, community and social enterprise (VCSE) sectors, employers and individuals together around common priorities and set out actions that can be taken to reduce the suicide rate over the next 5 years, improve | Bereavement support training for Police, Police and other stakeholders to explore opportunities for improving data collection and data sharing in all areas and to identify opportunities to improve the quality of intelligence and data that is used to improve our knowledge. | Capacity Building and Training-Oriented Engagements, Surveillance, Reporting and Data Systems Role |

(*Continued*)

| References | Geographical Scope | Scope of document/objective | Key Roles and Responsibilities of LEAs | Thematic Synthesis |
|---|---|---|---|---|
| | | support for people who have self-harmed and people bereaved by suicide | | |
| (The Youth Suicide Prevention Life Promotion Collaborative, 2023) | National Guidance | This resource is a guide for community and education-based providers that support youth and their families bereaved by suicide. | Police's role in postvention - securing the scene of death after the suicide has occurred, it is important for police and other supporting services to have good inter-sectoral collaboration. | Intersectoral Collaboration and Community Engagement, Strategic and System-Embedded Roles of Law Enforcement Agencies |
| (West Sussex County Council, 2023) | County Policy | This framework and action plan aim to reduce the risk of suicide in West Sussex. It aims to reduce the suicide rate over the next five years, with initial reductions observed within half this time or sooner, improve support for people who have self-harmed and improve support for people bereaved by suicide. | Real-time surveillance data has been gathered by police, training on suicide prevention across multiple agencies, including first responders such as police | Capacity Building and Training-Oriented Engagements, Surveillance, Reporting and Data Systems Role |
| (U.S. Department of Health and Human Services (HHS), 2024) | National Policy | To present priority actions the federal government will carry out in 2024 to 2026, to advance the goals and objectives of the 2024 National Strategy for Suicide Prevention. The Federal Action Plan represents commitments from 20+ federal agencies across 10 departments to support concrete actions for community-based prevention, treatment and crisis services, surveillance/quality improvement/research and health equity in suicide prevention. This is the first-ever Federal Action Plan accompanying the National Strategy. | Increase local collaboration and coordination between police, emergency services and other behavioural health crisis services to improve the quality of care | Intersectoral Collaboration and Community Engagement |
| (Surrey Suicide Prevention Partnership, 2025) | County Policy | Sets out an approach to reducing suicide in Surrey based on national and local intelligence/evidence, local learning and national suicide prevention recommendations. | Sets up the suspected suicide real-time surveillance database, part of the multi-agency audit focusing on Children and Young People aged 11 to 19-years-old who survived a suicide attempt within the last 6 months, crisis response for suicide, builds awareness of the links between domestic abuse and suicide within the force and partners with other sectors for the same. | Intersectoral Collaboration and Community Engagement, Surveillance, Reporting and Data Systems Role, Strategic and System-Embedded Roles of Law Enforcement Agencies |
| (Blackpool Council, 2025) | Local (sub-national) policy | Highlights potential modifiable risk factors from national and local data, cross-sector resources and support for mental health provided by statutory and non-statutory organisations in this area to: Improve mental health awareness and combat stigma, collaborate across the partnership and Blackpool communities, support anyone affected by suicide and build suicide-safer communities. | Emergency response service for suicide is a multi-agency partnership in real-time surveillance. | Surveillance, Reporting and Data Systems Role, Intersectoral Collaboration and Community Engagement |
| (Government of Sikkim, 2025) | State Policy | To guide policies, programmes and activities across the following strategic areas of action: a. Strengthening governance and leadership for mental health and suicide prevention, b. Ensuring comprehensive, integrated and community-based mental healthcare, c. Promotion of mental health and suicide prevention, d. Evidence-based suicide prevention interventions, e. Prevention, treatment and recovery for substance-use, alcohol and other addictions, f. Research for mental health and suicide prevention | Gatekeeper training programmes for first responders, including police | Capacity Building and Training-Oriented Engagements |

**Table 3.** (*Continued*)

| References | Geographical Scope | Scope of document/objective | Key Roles and Responsibilities of LEAs | Thematic Synthesis |
|---|---|---|---|---|
| (Hertfordshire Public Health Service and Hertfordshire County Council, 2025) | County Policy | sets out the local priority areas of focus identified through national and local data and intelligence and residents' insights | Real-time suicide surveillance, intersectoral sharing of suicide data | Surveillance, Reporting and Data Systems Role Intersectoral Collaboration and Community Engagement |
| (National Mental Health Services, 2025) | National Policy | This strategy commits to a whole-of-government approach towards suicide prevention, understanding the evolving facade of the national population and its implications for mental healthcare, keeping in context our increasing multiculturalism and varied post-COVID–19 effects | Gatekeeper in suicide prevention, improve collaboration between NMHS and the police force, police should be equipped to engage with, support, and transfer persons in crisis in a safe, respectful and dignified manner and in their support of the person's relatives, Regular police patrols at hotspots | Intersectoral Collaboration and Community Engagement, Strategic and System-Embedded Roles of Law Enforcement Agencies |
| (National Suicide Prevention Office, 2025) | National Policy | This Strategy outlines the actions required to realise a comprehensive approach to suicide prevention, aligning national efforts with the latest evidence and insights about what works. | Part of a co-response model that provides a rapid and tailored response to people in suicidal crisis | Intersectoral Collaboration and Community Engagement |
| (Public Health Jersey, 2025) | National Policy | Aims to prevent and reduce suicide in Jersey, to eliminate suicide by: a. Ensuring appropriate support is available to those at risk of suicide, and those affected by it b. Ensuring effective system coordination and joint working to prevent suicide c. Ensuring the availability of data, monitoring and ongoing learning over time d. Strengthening and supporting workforce capability and confidence in relation to suicide prevention across the Jersey system | Record all calls where a suicidal concern has been identified; use of social media by police in the identification of individuals at risk; referral in partnership with the Crisis Team, part of the crisis service for suicide incidents | Surveillance, Reporting and Data Systems Role, Intersectoral Collaboration and Community Engagement, Strategic and System-Embedded Roles of Law Enforcement Agencies |

to suicidal crises among LEA (Arensman et al., 2016; Marzano et al., 2016; Ko et al., 2021).

Out of 30, three resources described, mental health sensitisation training, which further defined LEAs' responsibilities in suicide prevention (Blais and Brisebois, 2021; Yeung et al., 2022; Roos and Fjellfeldt, 2023). Training included legal mandates governing LEA intervention, procedures for engaging mental health specialists in suicide risk assessment and referral and criteria for involuntary hospital transport (Yeung et al., 2022). Some programmes framed suicide awareness training within a broad, multi-level prevention approach, positioning law enforcement personnel as actors across individual-, subgroup- and population-level prevention activities (Roos and Fjellfeldt, 2023). In contrast, other training initiatives adopted a narrower focus, emphasising awareness of the consequences of not involving mental health specialists when responding to suicide-related behaviours and prioritising timely referral over wider preventive responsibilities (Blais and Brisebois 2021).

Across two studies, Crisis Intervention Training (CIT) was reported as the main framework guiding LEAs' responses to acute suicidal crises (Stokoe and Sikveland, 2019; Young et al., 2022). The CIT framework provides LEAs with structured mechanisms to recognise imminent suicide risk during emergency responses and engage individuals threatening, attempting or demonstrating imminent risk of suicide (Stokoe and Sikveland, 2019; Young et al., 2022). LEA's role through CIT is defined as identifying suicide risk indicators, establishing communication for de-escalation and distinguishing between ambivalent suicidal crises and suicide-by-cop situations (Stokoe and Sikveland, 2019) and facilitating referral to mental health emergency services rather than arrest or lethal force (Young et al., 2022).

De-escalation approaches described in the literature included the use of communication techniques, tactical positioning and psychological strategies to reduce threat while enabling life-saving intervention without lethal force (Stokoe and Sikveland, 2019), as well as specialised negotiation units employing team-based approaches in which primary negotiators engaged individuals in crisis through sustained dialogue, supported by secondary negotiators providing tactical guidance (Young et al., 2022). The literature additionally described bereavement support training that positioned law enforcement personnel as first points of contact following suicide deaths, with responsibilities for providing immediate, compassionate support and referral to bereavement services (Department of Health and Social Care, 2023).

Capacity-building and training-oriented engagement was the most frequently represented domain in peer-reviewed literature ($n = 28$), with studies largely employing quantitative, intervention-focused designs. Reported outcomes were primarily process indicators, including numbers trained, completion rates, programme duration and intensity and training infrastructure development. Individual-level outcomes (e.g., changes in knowledge, attitudes, confidence or preparedness) were less frequently reported, while organisational indicators (e.g., standard operating procedures, train-the-trainer models or formal linkages with mental health services) and population-level outcomes (e.g., identification and referral of suicidal individuals or crisis resolution trends) were rarely measured using standardised approaches. Effectiveness was typically inferred from self-reported or short-term pre–post findings from pilot or feasibility studies.

Training evaluations consistently showed uneven coverage but cognitive and attitudinal gains. Only 47.7% of officers in South Korea (Ko et al., 2021), 64% in the US Mountain West (Osteen et al., 2020) and between ~10% and ~ 74% across OSPI-Europe regions

had received suicide prevention training (Arensman et al., 2016), despite 82% reporting workplace exposure to suicidal individuals and 65% perceiving such encounters as likely. Training was associated with significant improvements in attitudes, including rejection of maladaptive beliefs in South Korea ($p \leq 0.035$), reductions in negative attitudes in the US Mountain West study (25.36 → 22.14; $p = 0.002$) and changes in stigma-related scores in OSPI-Europe (32.99 → 35.18; $p < 0.001$). Knowledge gains were robust across studies (e.g., UK: 6/10 → 8/10; $p < 0.0001$, OSPI-Europe: 3.88 → 4.35; $p < 0.001$, US Mountain West (Osteen et al., 2021): 5.10 → 5.79; $p < 0.001$, COPS (Hofmann et al., 2021): 22.44 → 26.49; $p < 0.001$), though objective knowledge did not improve in Nigeria (5.5 → 5.5; $p = 0.8$). In another scenario-based study, 76.2% of scenarios were terminated by instructors, 32.1% ended with firearm removal and 27.8% resulted in the subject being shot (Young et al., 2022).

### Surveillance, reporting and data systems role

A total of 30 literary sources described the role of LEA in suicide surveillance and the use of this surveillance data in suicide prevention. Out of the 22 included sources, 16 were peer-reviewed literature.

Across the literature, law enforcement agencies were positioned as central actors in suicide surveillance due to their statutory responsibility to attend sudden and unexpected deaths. This role placed them as primary custodians of suicide-related data, contributing to early detection, prevention and postvention within national suicide prevention systems.

LEA-led surveillance was described as involving the systematic recording of suspected suicide deaths at or near the time of occurrence, capturing demographic characteristics, circumstances of death, methods used and contextual information, enabling near real-time monitoring of suicide trends.

The literature also described the use of surveillance data to identify high-risk locations, informing targeted, location-based prevention efforts and to facilitate postvention through referral of bereaved individuals to support services following consent at the scene. In addition, law enforcement agencies were reported to control the initial flow of suicide-related information, including classification of suspected suicides, custody of investigative documentation and transmission of information to health systems and policymakers, with sole authority to declare suicide as a cause of death in several settings (Hagaman et al., 2016; Ullah et al., 2021).

Evidence for surveillance, reporting and data systems roles was drawn from a smaller body of studies ($n = 16$) describing LEA-led or LEA-involved early alert, real-time reporting and data-linkage initiatives. These studies emphasised system development and implementation processes, with indicators focused on reporting mechanisms, interagency data linkage, timeliness of identification and information flow to prevention or postvention actors. Organisational indicators such as governance arrangements, data-sharing protocols and role operationalisation were commonly noted, while effectiveness was inferred from perceived improvements in situational awareness or coordination rather than from controlled or longitudinal evaluations.

In the UK, a police-led suicide surveillance strategy captured 78.8% of suspected suicides via police notification of death (NoD) forms, compared with 94.2% captured through coroner reports (McGeechan et al., 2017), while a similar study elsewhere showed 98% population coverage, with ~98% data completeness, enabling identification of clusters and high-risk locations (Marzano et al., 2023). Among NoD cases, 88% increase in referrals compared with

the previous year, with approximately 75% of referrals originating from police NoD forms.

### Intersectoral collaboration and community engagement

This section synthesises findings from 38 sources examining LEA roles in intersectoral collaboration and community engagement for suicide prevention, comprising 17 peer-reviewed studies and 21 grey literature sources.

Across the literature, law enforcement agencies were consistently positioned as key partners in intersectoral suicide prevention efforts, reflecting their role as first responders and their access to population- and situation-level data at local, regional and national levels. Their involvement was described as spanning information sharing, participation in real-time suicide surveillance systems and coordination of early response and postvention activities, including timely communication of suspected suicides to public health and community partners to support identification of clusters, locations of concern and emerging risks. A few studies reported fragmented collaboration and constraints on prevention planning, where suicide data held by law enforcement were not effectively shared with health systems, (Dawson et al., 2021).

The literature also described collaborative service delivery models, including co-response arrangements in which law enforcement worked alongside mental health professionals to manage suicide-related crises and facilitate diversion to appropriate care pathways. Beyond health sector collaboration, law enforcement engagement extended to coordination with transport authorities, housing, education and social services to support means restriction initiatives, targeted prevention at high-risk locations and secondary prevention activities.

In addition, law enforcement agencies were reported to engage in community-facing prevention, including participation in suicide prevention campaigns, dissemination of information on support services and collaboration with community organisations to reduce stigma and promote help-seeking (Decker et al., 2018; Jager-Hyman et al., 2019; Bland et al., 2021; Ullah et al., 2021). Across several sources, law enforcement personnel were positioned as intermediaries linking individuals, families and communities to suicide prevention and bereavement support resources.

Intersectoral collaboration and community engagement were examined in 17 studies describing coordination between LEAs, health services, community organisations and policy actors. Indicators centred on formal linkage agreements, joint or co-responder models, referral pathways, shared protocols and information exchange, alongside community outreach and stakeholder participation. Effectiveness was rarely assessed directly and was instead inferred from stakeholder-reported benefits, improved coordination or descriptive changes in service engagement within community-based programmes.

Joint police–mental health mobile response unit recorded 296 crisis contacts; one in three encounters addressed suicide risk (Lee et al., 2015). The Swedish Psychiatric Emergency Response Team (PAM) handled 1,580 service requests, with 80% resulting in attended cases; multi-agency involvement was common, with ambulance services engaged in 55%, police in 49% and rescue services in 7% of cases and only 24% managed by PAM alone (Bouveng et al., 2017). Canadian co-response police–mental health programme demonstrated statistically significant system-level improvements, including reductions in use of force (ATE = −0.077; $p \leq 0.05$) and hospital transports (ATE = −0.773; $p \leq 0.01$), alongside increases in community referrals (ATE = 0.285; $p \leq 0.01$) and management through social networks (ATE = 0.530; $p \leq 0.01$) (Blais and Brisebois, 2021).

### Means restriction and environmental prevention roles

This section synthesises findings from seven included sources, three peer-reviewed and four grey literature documents, examining the role of LEA in implementing means restriction as a suicide prevention strategy.

Across included sources, means restriction was consistently identified as an effective suicide prevention strategy, with LEA positioned as key operational actors due to their legal authority, first-responder role and presence across community, custodial and environmental contexts. Their involvement was most frequently reported in relation to firearm access restriction, particularly through temporary, voluntary firearm storage during periods of elevated suicide risk (Runyan et al., 2017; Brooks-Russell et al., 2019; Illinois Department of Public Health, 2019; The Mental Health Services Oversight and Accountability Commission, 2019). LEA also routinely advised removal of firearms from homes during crises and promoted secure storage practices, including locked safes, trigger locks and separate ammunition storage. Reported barriers included limited storage capacity, liability concerns and legal uncertainty, with variation across jurisdictions reflecting differences in firearm legislation (Runyan et al., 2017; Brooks-Russell et al., 2019).

The reviewed resources also documented the role of LEAs in means restriction at identified suicide hotspots, where they supported environmental interventions, such as barriers, fencing and controlled access, alongside routine and targeted patrolling, active surveillance and rapid response protocols to enable timely intervention. (Brooks-Russell et al., 2019). In domestic and acute crisis contexts, they facilitated the temporary removal of lethal means, particularly firearms, using discretionary powers or legal mechanisms such as protection orders. Within custodial settings, means restriction was operationalised through environmental design and supervision practices aimed at reducing access to ligature points and hazardous materials (National Human Rights Commission India, 2014).

Means restriction and environmental prevention roles were least represented, with only three peer-reviewed literature examining LEA involvement in hotspot interventions and temporary firearm storage initiatives. Indicators were predominantly process-oriented, including environmental modifications, partnership development and availability or uptake of storage options. Individual- and population-level outcomes, such as changes in safe storage behaviours or suicide attempts at hotspots, were infrequently reported and lacked standardised measurement. Population-level outcomes were observed in Australia at a jumping site, where female suicides showed a significant upward trend pre-intervention (APC = 16.64%, $p < 0.001$), followed by a significant post-intervention decline (APC = −21.27%, $p = 0.01$), while male suicides showed a non-significant upward trend (APC = 6.23%, $p = 0.06$) (Ross et al., 2020).

### Discussion

This scoping review systematically mapped and examined the global evidence on LEAs and suicide prevention. It also synthesised the temporal, geographical and structural variation of suggested functional roles and responsibilities across the extracted evidence. The findings of the review demonstrate increasing recognition of LEAs as an important stakeholder in suicide prevention in the high-income countries (Marzano et al., 2016; Florida Department of Children and Families, 2023; Garratt et al., 2023; Roos and Fjellfeldt, 2023), where their roles have moved beyond the classical policing towards preventive and supportive functions over time. Conversely, in LMICs,

the evidence (Arya, 2024; Ministry of Health and Family Welfare 2022; Ministry of Health Malaysia, 2013; National Human Rights Commission India, 2014; Olibamoyo, 2023; Royal Government of Bhutan, 2018) is still evolving and limited.

### Structural and institutional constraints shaping law enforcement roles in suicide prevention

The evidence base on the role of LEAs in suicide prevention has evolved unevenly over time. Early literature from the 1960s to the 1990s was limited and largely descriptive, framing suicide primarily as a law-and-order issue and positioning LEA as investigators or custodians of legal process (McGee, 1968; Olivero and Hansen, 1994; Linsley et al., 2007). A gradual shift emerged with the development of global mental health and suicide prevention frameworks, including the WHO's Comprehensive Mental Health Action Plan and the Sustainable Development Goals, the *LIVE LIFE* implementation guide, which reframed suicide as a public health concern and recognised the need for multisectoral engagement (World Health Organization, 2009; United Nations, 2015a, 2015b). This shift, reinforced by policing reforms in many HICs, was accompanied by an expansion of LEA roles in policy and practice and a growing body of empirical research examining police involvement in suicide prevention. However, this evolution has been geographically uneven. While evidence from HICs has expanded, contributions from LMICs remain sporadic (Williamson, 2008).

This pattern reflects deeper inequities in research priorities, including limited domain specialisation, weaker systemic research support and constrained funding availability in many LMICs. Suicide research in these settings has remained disproportionately focused on health system strengthening (Vijayakumar and Phillips, 2016; Jacob, 2017), in contrast to high-income countries. As a result, the role of non-health system actors, particularly LEAs have received limited and inconsistent attention. These research gaps are closely intertwined with the historical criminalisation of suicide across many LMICs. For decades, colonial-era legal frameworks constructed suicide as a violation of law, positioning LEAs primarily as agents of investigation (Ochuku et al., 2022; Palit, 2024). Although suicide has now been formally decriminalised in most LMICs, this transition has been gradual and uneven. Crucially, legal reform has not been matched by commensurate changes in field-level implementation, institutional mandates, or behavioural norms within policing systems (Gupta, 2024). Standard operating procedures, training curricula and accountability mechanisms have often remained unchanged, allowing legacy interpretations of suicide to persist in practice. As a result, law enforcement engagement with suicide in many LMICs continues to be largely reactive and procedural, focused on documentation, examination and reporting rather than prevention, early intervention, or facilitation or support for care. This institutional positioning is reinforced by enforcement-oriented policing models that leave limited space for community-based or preventive roles. Unlike HICs, where sustained reforms have institutionalised community policing, trauma-informed training and partnerships with health and social services, comparable transformations within LMIC law enforcement systems remain limited (Bott et al., 2005; Arya et al., 2023).

Within this context, the role of law enforcement in suicide prevention has remained poorly conceptualised and inconsistently examined. The scarcity of locally generated evidence has produced a self-reinforcing cycle: in the absence of structured frameworks and empirical evaluation, preventive and facilitative functions remain ambiguous; this ambiguity limits inclusion of LEAs in formal suicide prevention strategies, reduces recognition of their public-facing and first-contact role and constrains investment in generating evidence on what works, under what conditions and to what extent. Consequently, the lack of evidence both reflects and perpetuates the marginalisation of intersectoral actors in suicide prevention.

### From legal reform to preventive practice: emerging opportunities and persistent gaps

Recent legal and policy developments nevertheless create openings for reconfiguration. Various emerging global policy frameworks increasingly emphasise rights-based, trauma-sensitive and empathetic responses to individuals at risk of suicide and other vulnerable populations. Similarly, the Mental health reforms that are happening across various LMICs increasingly frame suicide as a public health issue and promote multisectoral collaboration (Vijayakumar and Phillips, 2016). These principles are particularly salient in LMIC contexts, where LEAs are often the first and sometimes the only institutional actors to engage with individuals during acute mental health and suicide crises (van der Feltz-Cornelis et al., 2011). Such encounters raise significant ethical and operational dilemmas, including risks of escalation, excessive use of force and phenomena such as "suicide by cop," underscoring the importance of de-escalation and harm minimisation (Lamb et al., 2002; Police Executive Research Forum, 2019).

However, legal reform alone is insufficient without corresponding changes in institutional capacity-building training, practice, institutional incentives and evidence generation (Lamb et al., 2002). Task sharing must be supported through sustained training, organisational reform and cultural change within policing institutions and structured integration of law enforcement into community-based suicide prevention systems (Dorji et al., 2017). When grounded in local socio-legal and cultural realities, such approaches offer the potential to shift policing from a reactive, event-driven orientation towards preventive and collaborative engagement.

### Strength and limitations

The findings of this scoping review should be seen considering some methodological limitations. We specifically examined selected databases and limited search across search engines, however, there remains the possibility that this review did not capture the full range of literature available in this intersection, particularly non-English publications and grey literature not available in open domain. Due to time and resource constraints, the websites of individual governmental and non-governmental organisations could not be accessed to retrieve relevant policy or guidance documents, among others. The review process did not aim to assess their methodological quality, as it was outside the scope and the heterogeneous nature of the available literature, and therefore, the strength of the evidence cannot be established. Nonetheless, to the best of our knowledge, this is the first scoping review that systematically examines the role of LEA in suicide prevention, providing an initial foundation for understanding how this sector has been positioned within the broader public health discourse and highlighting directions for future, nuanced research for various sub-domains.

### Future research

Future research should aim to a focus on transferable practices and models (strategies, directives and interventions employed or suggested for LEA) from other settings. Further research should also

test evidence-based practices such as gatekeeper training, community policing partnerships and social service linkages involving LEA engagement within a systematic, multisectoral context. Most importantly, such research studies must examine the structural, legal and cultural factors that shape LEAs' roles, as these are likely to either enable or hinder their effectiveness. More rigorous designs, such as randomised controlled trials, longitudinal studies and mixed-method designs, are required to assess both proximal effects (e.g., knowledge, attitude, support provision, stigma and referral behaviour) and distal effects (e.g., suicide attempts, deaths). Strengthening the evidence base will not only guide policy but also inform the adaptation of global best practices to the realities of LMIC settings beyond the conventional law enforcement function towards more comprehensive, community-based public health functions.

## Conclusion

This scoping review synthesises the developing body of literature on the role of LEA in preventing suicide to fill an essential knowledge gap in scholarly research and policy discussion. The review highlights substantial heterogeneity and sparse evidence from LMICs and a dearth of evidence-based, structured, scalable models that aim to prevent suicide. This highlights the pressing necessity for context-informed, cross-sectoral models informed by experience, evidence-based and integrated into national suicide prevention strategies in low-resource settings with high suicide burden.

**Open peer review.** To view the open peer review materials for this article, please visit http://doi.org/10.1017/gmh.2026.10186.

**Supplementary material.** The supplementary material for this article can be found at http://doi.org/10.1017/gmh.2026.10186.

**Data availability statement.** The authors confirm that the data supporting the findings of this study are available within the article and its supplementary material.

**Author contribution.** Nikhil Jain and Isha Lohumi conceptualised the research and developed the search strategy. Nikhil Jain, Isha Lohumi and Aratrika Datta ran the search strategy. Nikhil Jain, Isha Lohumi, Aratrika Datta and Niranjana Regimon conducted de-duplication and screening, while Isha Lohumi and Niranjana Regimon carried out data extraction. Nikhil Jain, Isha Lohumi, Niranjana Regimon and Aratrika Datta prepared the first draft of the manuscript. All authors contributed to manuscript editing and revisions, and all authors reviewed and approved the final version of the manuscript.

**Financial support.** The study is a part of the STRIDE action research project, funded by the Mariwala Health Initiative (India) and implemented by the Indian Law Society, Pune. Open access funding provided by Maastricht University.

**Competing interests.** The authors declare none.

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
