## [Reviewer Report]

This manuscript addresses a very important topic of the suicide prevention and Law Enforcement Professionals (LEP), and it surely contributes to the new insights on suicide prevention concerning LEP. However, several methodological issues limit the interpretability and generalisability of the findings.

- Page 8: You mention 1243 studies from 7 databases were included, and it is unclear if so many could be found on the topic of both LEP and suicide or suicide prevention. More information on a number and kind of articles/texts identified in these specific databases would enrich this review.

- Page 28: Fig 1 Types of Evidence Identified in the Scoping Review - it would be valuable to add in the workflow even numbers of texts analysed.

- Page 33: Table No. 1. Intervention-Based Studies. It would be important to add even size of the samples included in these studies. Finally, the inclusion of participant flow details (e.g., numbers invited, enrolled, and retained) and a timeline of the intervention would improve transparency and facilitate replication. Reporting these aspects follows best practices outlined in CONSORT and PRISMA guidelines for educational and intervention research.

---

## [Reviewer Report]

The manuscript provides a narrative review of the role of law enforcement personnel in suicide prevention. Given the role of law enforcement personnel in responding to suicidal calls, it is important to understand their impact on these calls, which I am not aware of studies that have examined this. While the topic is timely and relevant, and the manuscript has potential to make a contribution by synthesizing studies in this area, the manuscript has several limitations that prevent such contribution. Overall, the findings lack depth and detail on the themes found in the literature, with few examples of passages from the found literature to support the “themes.” For example, some of the results for themes are one to two sentences. I also put themes in quotes because in quotes because the narrative review is not organized by themes, but rather by the type of article such as peer-reviewed articles, guidance documents, and policies. The authors find similar themes across the types of articles, which are mentioned in their abstract—evaluation of training, capacity building, restricting means, surveillance initiatives, etc., so I would recommend the authors reorganize the findings around these substantive themes. As it reads now, it is a bit redundant because these themes get repeated across the different types of articles. Additionally, some of the themes have little to do with law enforcement personnel such as restricting means. The authors should incorporate more language and examples from the sources to highlight how law enforcement fits (or not) into these topics. As the review stands now, it makes a nice reference guide on how countries view the role of law enforcement in suicide prevention, but lacks substance.

The title implies the focus will be on law enforcement personnel when responding to suicides, but the review turns out to be more focus on policies and guidelines rather than personnel, so the title is a bit misleading. The keywords capture this, so perhaps remove “personnel” from the title. There are two areas that pertain to personnel with intervention studies, but that is only one topic/area of articles. The unit of analysis in the various guidelines or “resources” as the authors refer to, is more at an agency level and some at the national level.

Perhaps the largest weakness of the study is the limited number of search terms for the review, particularly only using “law enforcement personnel” and not including “police,” “police officers,” “policing” as search terms. This likely excluded many studies that may examine police and suicide responses.

How were the non-intervention studies different from the intervention studies and the grey literature? Again, reorganizing the results around the substantive themes may help address this.

The authors mention RCTs in the discussion on page 17 and how these are limited in LMICs, but do not discuss any methodologies of the articles in the review. Were any of the interventions studies RCTs, were there differences by HICs and LMICs in the methodologies of these interventions, such as the use of RCTs? The authors need additional support from their review or other research to make this claim.

The authors refer to “recommended evidence-based approaches” on page 18, line 579-580, but I would argue these are not evidence-based as there are few rigorous studies on these trainings/responses. Perhaps “best-practices” is a better term.

The authors argue that in LMICs, LEPs are the “sole and sometimes sole point of institutional engagement during acute crisis” (pg 18, line 591), but I would argue that for much of the U.S. that is still the case.

The authors bring up other topics in the discussion on page 18, lines 592-593, as “literature identifies.” Were these found in the current review, or are the authors referring to other literature not discussed in the review?

More minor issues:

Move Figure 2 to before section 2.3 after it is referred to at the end of section 2.2.

The second paragraph of the results section fits more as “data analysis;” instead, the authors should tailor this paragraph to how the results will be presented, first chronologically and then by type (or theme based on my comments earlier).

Not sure “evidence” is a fitting term for the results section 3.1. The resource and guideline documents found in the search do not suggest evidence on law enforcement role in suicide prevention. Evidence sounds like studying law enforcement impacts of suicide prevention through evaluation studies, which is not what the results provide.

The authors use the acronyms HICs and LMICs in the abstract , but only define HIC and do not define either in the manuscript, only using the acronym. The authors should spell out the acronym in the manuscript the first time it is used.

Some grammatical errors throughout so the manuscript needs some careful editing.

Pg. 4, line 107- change “there” to “their”

Pg. 7, line 229- suggest adding “identifying” before “the role of law enforcement”

Pg. 15, line 473- change “focussed” to “focused”

Pg. 17, line 570- remove comma after “institutionalizing”

Pg. 18, line 578- “These comprises” needs fixed

---

## [Editor Report]

Dear Prof Jain,

Thank you for your submission. Given that suicide has strong links in the criminal justice system, including police, from investigating a suicide to ensuring the safety of individuals who are at risk, this is a much needed scoping review. 

The reviewers raise some valid points, including but not limited to the limited number of keywords used in the search. Please address them carefully and respond to each point raised.

Furthermore, one point that would strengthen the manuscript is a discussion which contextualizes the findings and law enforcement personnel in larger ecosystems of regional or national strategies. It would be good to know if substantive changes needed to be made to optimize law enforcement for suicide prevention ecosystems.

I look forward to reading the revised manuscript.

Thank you and all the best,

Dr Sandersan Onie

---

## [Reviewer Report]

The manuscript has greatly improved with the reorganization of the findings based on thematic topics found in the studies. This helped streamline the manuscript and the findings are more informative. The authors also provided clarification to the methodology that strengthens the manuscript. I don’ think it is necessary to include in-text citations for every article that fits a characteristic such as peer-reviewed or grey literature on page 8. This page (as well as page 10 and19) are mostly citations, and that is what the table is for. If you want to highlight something specific about a particular study, then cite it, but citing everything seems unnecessary. It will shorten the main text of the manuscript, but again, readers can refer to the tables for which studies are different types and discuss the themes. The tables and figures (and appendices) are not labeled, so it is difficult to identify the appropriate table or figure that is referenced in the text. Other than these changes, I think the manuscript is suitable for publication.

---

## [Reviewer Report]

Thank you for all the revisions: 1) reframing of the unit of interest from “personnel” to law enforcement agencies, 2) expanding and documenting the search approach with explicit inclusion of police terminology, and 3) strengthening the Results narrative with clear thematic structure and more concrete examples of what LEAs do within training, crisis response, surveillance, and postvention models. This revision meaningfully improves the manuscript’s alignment with its stated purpose and with global mental health priorities.

---

## [Editor Report]

Dear Dr Jain,

Thank you for the comprehensive revisions to your manuscript. Please address the review comments carefully and I look forward to receiving a revised manuscript.

Thank you and all the best,

Dr. Sandersan Onie

---

## [Editor Report]

Dear Dr Jain,

I am now pleased to recommend this manuscript for publication. Thank you for your extensive work on this manuscript. I am sure this will be an important contribution to the suicide prevention literature.

Thank you for continuing work on this.

All the best,

Dr. Sandersan Onie